# Fast retrieval of XCO$_2$ over East Asia based on the OCO-2 spectral measurements

Fengxin Xie[1,a], Tao Ren[1], Changying Zhao[1], Yuan Wen[2], Yilei Gu[2], Minqiang Zhou[3], Pucai Wang[3], Kei Shiomi[4], and Isamu Morino[5]

[1]China-UK Low Carbon College, Shanghai Jiao Tong University, Shanghai, China
[2]Shanghai Institute of Satellite Engineering, Shanghai, China
[3]Institute of Atmospheric Physics, Chinese Academy of Science, Beijing, China
[4]Earth Observation Research Center, Japan Aerospace Exploration Agency, 2-1-1 Sengen, Tsukuba, Ibaraki 305-8505, Japan
[5]Satellite Remote Sensing Section and Satellite Observation Center, Earth system Division, National Institute for Environmental Studies, Onogawa 16-2, Tsukuba, Ibaraki 305-8506, Japan
[a]Currently at: Atmosphere and Ocean Research Institute, the University of Tokyo, Chiba, Japan

**Correspondence:** Tao Ren (tao.ren@sjtu.edu.cn)

**Abstract.** The increase in greenhouse gas concentrations, particularly CO$_2$, has significant implications for global climate patterns and various aspects of human life. Spaceborne remote sensing satellites play a crucial role in high-resolution monitoring of atmospheric CO$_2$. However, the next generation of greenhouse gas monitoring satellites is expected to face challenges, particularly in terms of computational efficiency in atmospheric CO$_2$ retrieving and analysis. To address these challenges, this study focuses on improving the speed of retrieving the column-averaged dry air mole fraction of carbon dioxide (XCO$_2$) using spectral data from the OCO-2 satellite, while still maintaining retrieval accuracy. A novel approach based on neural network (NN) models is proposed to tackle the nonlinear inversion problems associated with XCO$_2$ retrievals. The study employs a data-driven supervised learning method and explores two distinct training strategies. Firstly, training is conducted using experimental data obtained from the inversion of the operational optimization model, which is released as the OCO-2 satellite products. Secondly, training is performed using a simulated dataset generated by an accurate forward calculation model. The inversion and prediction performance of the machine learning model for XCO$_2$ are compared, analyzed, and discussed for the observed region over East Asia. The results demonstrate that the model trained on simulated data accurately predicts XCO$_2$ in the target area. Furthermore, when compared to OCO-2 satellite product data, the developed XCO$_2$ retrieval model not only achieves rapid predictions (<1 ms) with good accuracy (1.8 ppm or approximately 0.45%), but also effectively captures sudden increases of XCO$_2$ plumes near industrial emission sources. The accuracy of the machine learning model's retrieval results is validated against reliable data from TCCON sites, demonstrating its capability to capture CO$_2$ seasonal variations and annual growth trends effectively.

## 1 Introduction

Since the Industrial Revolution, human activities have released large amounts of greenhouse gases, primarily carbon dioxide, into the atmosphere. This continual increase in emissions has led to global warming and disrupted human societies and ecosys-

tems (Zehr, 2015). Accurately estimating atmospheric carbon fluxes is critical for implementing effective emission reduction strategies at national and regional levels. However, precise carbon flux estimates require assimilating carbon dioxide concentration data across regions, using measurements of atmospheric column-averaged dry air mole fraction of carbon dioxide ($XCO_2$) (Jin et al., 2021). Direct measurement methods like balloons or aircraft have challenges in obtaining global-scale data. Currently, the main monitoring approach uses spectrometers to record spectra in $CO_2$ absorption bands, followed by inversion algorithms to derive $XCO_2$. The two primary monitoring methods are ground-based monitoring stations and satellite remote sensing.

The Total Carbon Column Observing Network (TCCON) provides ground-based monitoring of atmospheric carbon dioxide through a global network of high-precision Fourier transform spectrometers (Wunch et al., 2011, 2015). However, TCCON sites are sparsely distributed and cannot be deployed in regions with unfavorable geography or harsh climates. Consequently, the network lacks the extensive spatial coverage required for comprehensive global carbon monitoring and carbon cycle analysis. Nevertheless, the ultra-high spectral resolution of TCCON spectrometers enables highly accurate retrievals of $XCO_2$. Under clear sky conditions, TCCON precision can reach 0.1% (<0.4 ppm). Under relatively clear conditions with minimal clouds and aerosols, precision remains within 0.25% (<1 ppm) (Messerschmidt et al., 2011). Due to such high precision and accuracy, TCCON data are invaluable for validating satellite-based $XCO_2$ products (Cogan et al., 2012; Wunch et al., 2017; Liang et al., 2017) and comparing them to carbon cycle models. However, the spatial limitations of the network underscore the need for satellite remote sensing to provide regular global measurements of atmospheric carbon dioxide.

High-spectral-resolution greenhouse gas monitoring satellites employ spectrometers on orbit to measure solar radiation spectra after interaction with the Earth's atmosphere and ground surface (Meng et al., 2022). Unlike ground monitoring, satellite remote sensing offers broader spatial coverage and more flexible temporal observation globally. Consequently, satellite remote sensing has become vital for greenhouse gas monitoring worldwide. Notable ongoing passive $CO_2$ observation missions include China's TanSat (Liu et al., 2018), Japan's GOSAT (2009) and GOSAT-2 (2018) (Hamazaki et al., 2005; Kuze et al., 2009; Imasu et al., 2023), and the United States' OCO-2 (2014) and OCO-3 (2018) (Crisp et al., 2017; Eldering et al., 2019). Upcoming missions are France's MicroCarb by CNES (Cansot et al., 2023), ESA's $CO_2M$ (Sierk et al., 2021) and GOSAT-GW (Matsunaga and Tanimoto, 2022). The next-generation greenhouse gases monitoring satellites mainly address the challenge of improving the spatial and temporal resolutions of observations. However, single satellites still have resolution, coverage, and meteorological limitations for regional emission monitoring. Enhancing satellite sensor performance alone cannot produce datasets sufficient for monitoring carbon sources and sinks. Improving the accuracy and efficiency of satellite data inversion is also crucial. Integrating data from multiple satellites into a coordinated system is necessary to fully capture dynamic changes in regional carbon sources and sinks. Developing new high-precision, high-throughput inversion methods to efficiently derive accurate greenhouse gas concentration distributions from satellite data is a key challenge needing attention.

The mainstream inversion algorithms (O'Dell et al., 2012; Crisp et al., 2012; Yoshida et al., 2013) for retrieving greenhouse gas concentrations from high-spectral-resolution satellite measurements are based on nonlinear Bayesian optimization theory (Rodgers, 2000) and full physics models. In essence, these algorithms operate by iteratively adjusting estimated gas concentration profiles and other atmospheric and surface parameters in a radiative forward model to minimize the mismatch

between simulated and observed spectra. More specifically, the inversion process starts with an initial atmospheric state guess, including trace gas concentration profiles as functions of pressure/altitude. Radiative transfer equations are then solved to simulate the top-of-atmosphere radiance spectrum observed by the satellite for this atmospheric state. The simulated spectrum is compared to the actual observed spectrum, calculating the difference, covariance and cost function. The input gas profiles and model parameters are iteratively adjusted to reduce the cost function over multiple rounds of radiative transfer simulations. Once simulated spectra closely match observations, the model state is output as the retrieved concentration profile. However, executing these complex optimizations requires computationally expensive interpolation of high-spectral-resolution gas absorption reference data and solving the radiative transfer equations in each iteration. Running the radiative forward model repeatedly for every adjusted atmospheric state also leads to slow overall inversion. Consequently, optimization-based retrievals struggle to match increasing satellite observation volumes and throughput needs. This inherent inefficiency has become a major obstacle to operational greenhouse gas monitoring using current and planned high-resolution spectrometers. While rigorous, standard nonlinear optimization retrievals lack the speed and scalability required for high-precision real-time or near-real-time greenhouse gas mapping satellite-based greenhouse gas mapping. Overcoming this bottleneck necessitates new inversion approaches that can ingest high-resolution spectral data and retrieve concentrations with both accuracy and computational efficiency.

In recent years, machine learning has demonstrated exceptional performance across various research fields, with the discovery of potential nonlinear relationships between data as one of its fundamental and crucial applications. Regarding the important applications of carbon dioxide ($CO_2$) retrieving, Carvalho et al. (2010) attempted to retrieve the vertical $CO_2$ profiles using spectral data from SCIAMACHY's 6 channels (1000-1700 nm). The overall precision and bias of the retrieved results were estimated to be approximately 1.0% and less than 3.0%, respectively. Gribanov et al. (2010) developed a two-hidden-layer multilayer perceptron (MLP) model to retrieve $CO_2$ vertical concentrations by reflected solar radiation measured by the GOSAT TANSO-FTS sensor, achieving an inversion accuracy better than 1 ppm for $CO_2$ column-averaged values and better than 4 ppm for surface $CO_2$ concentrations for the test samples. In the study conducted by Zhao et al. (2022), a two-step machine learning approach was developed for retrieving atmospheric $XCO_2$ using spectral data from the GOSAT weak-$CO_2$ band. They established a direct one-dimensional line-by-line forward model to simulate GOSAT's observed spectra within the 6180-6280 $cm^{-1}$ spectral interval, forming the foundation for training their machine learning model. The retrieval model operates by initially obtaining the atmospheric spectral optical thickness, followed by extracting $XCO_2$ from this optical thickness spectra. As a proof-of-concept, the method was tested in Australia under clear sky conditions using GOSAT's spectra, demonstrating an accuracy of approximately 3 ppm for $XCO_2$ retrieval. The study also discussed potential enhancements to further refine the accuracy of this retrieval method. Keely et al. (2023) employed the machine learning method of Extreme Gradient Boosting (Chen and Guestrin, 2016) to develop a nonlinear bias correction approach for OCO-2 version 10 product, significantly reducing systematic errors in $CO_2$ measurements and improving data quality, with an increase in sounding throughput by 14%. David et al. (2021) and Bréon et al. (2022) attempted to establish correlations between $XCO_2$ in the European Centre for Medium-Range Weather Forecasts' CAMS (Copernicus Atmosphere Monitoring Service) database and OCO-2 satellite monitoring spectra using multilayer perceptron artificial neural network models. However, their recent research (Bacour et al.,

2023) indicates that when the test dataset extends beyond the time range covered by the training dataset, the predicted results show a slight bias, approximately 2.5 ppm per year. Practical deployment of machine learning techniques for remote sensing demands additional research into the generalization performance of models on new observational data distributions beyond those encountered during training.

In the present paper, a proof-of-concept study demonstrates a novel machine learning strategy to accurately and efficiently retrieve atmospheric $XCO_2$ value from OCO-2 satellite spectral measurements. The model rapidly retrieves $XCO_2$ directly from OCO-2 spectral data, eliminating the need for repetitive radiative transfer simulations required by traditional nonlinear optimization retrieval algorithms. Additionally, the model enables the prediction of future $XCO_2$ values. The method was validated by comparing the retrieved $XCO_2$ against OCO-2 satellite version-10r products and ground-based TCCON mea-

surements, confirming the accuracy of our efficient spectral inversion approach. The model also successfully demonstrated its capability to detect local plume features, indicating its potential utility in monitoring and analyzing specific emission sources. A major innovation in the present study is using accurate radiative transfer simulations to generate the training data, rather than relying solely on experimental data products. This simulation-based training approach could help overcome limitations in existing experimental data. Additionally, our neural network model achieves $XCO_2$ retrieval speeds orders of magnitude

faster than traditional methods, reducing computation time from multiple seconds to less than 1 millisecond. This dramatic improvement in retrieval efficiency could enable real-time processing of the massive data volumes expected from next-generation greenhouse gas monitoring satellites. Importantly, our model achieves a precision of less than 1.8 ppm, competitive with the current state-of-the-art in retrieval accuracy. We also demonstrate the ability to accurately capture temporal variations and trends in $XCO_2$ by validating against reliable TCCON ground-based data. This level of verifiable performance is an important

capability. This provides an effective solution for rapid inversion of large-scale, high-spectral-resolution remote sensing data in the future.

## 2   The machine learning based $XCO_2$ retrieval model

### 2.1   Targeted area and data screening

This proof-of-concept study aims to develop and validate an accurate and efficient machine learning-based $XCO_2$ retrieval

model applied to the long OCO-2 time series for the East Asian region. Currently, similar global $XCO_2$ retrieval models rely on computationally intensive physical models. Our goal is to demonstrate a more efficient data-driven approach using MLP neural networks.

    Before developing the machine learning based fast retrieval model, we implemented several preprocessing steps on the OCO-2 observational dataset (OCO-2 Science Team et al., 2020a) for the target East Asian area spanning between 20°N-

120 45°N and 110°E-145°E, as shown in Fig. 1. Specifically, we filtered the data both spatially and temporally to focus only on observations within this geographic region and time period of interest (2016-2021). Additionally, we filtered the data to only include Nadir mode observations marked as "Good" based on the quality flag indicator ("xco2_quality_flag" = 0 in OCO-2 Lite v10r files (OCO-2 Science Team et al., 2020b)), as these represent the highest quality OCO-2 measurements.

Several TCCON ground stations located in this region (e.g. Hefei, Saga, Tsukuba, Xianghe, Anmyeondo and Rikubetsu),
as shown in Fig. 1, provide valuable ground-truth $XCO_2$ data for validating the MLP model predictions. If the model can
accurately reproduce the TCCON observations from corresponding OCO-2 measurements, it suggests the model has learned
meaningful relationships between the satellite data and underlying $CO_2$ concentrations.

Furthermore, the successful demonstration of accurate $XCO_2$ retrieval over East Asia is a first step toward expanding this
approach globally. The model could be retrained or supplemented with additional regional data to extend coverage. By combin-
ing reliable regional MLP models, global $XCO_2$ maps could be retrieved. This "jigsaw puzzle" strategy would further validate
the feasibility of global-scale machine learning-based $XCO_2$ retrievals from satellite observations.

## 2.2 The artificial neural network architecture

This study introduces a multilayer perceptron (MLP) neural network model for estimating $XCO_2$ from OCO-2 satellite obser-
vations. Inspired by David et al. (2021) and Bréon et al. (2022), the "MLP-$XCO_2$" model input layer is designed based on the
measurement principles of OCO-2 and atmospheric radiative transfer effects on the observed spectra, the artificial neural net-
works architect is shown in Fig. 2. Specifically, the MLP model input layer consists of spectral information, surface pressure,
the corresponding year, and geographical observation information as summarized in Table 1 and explained below.

**Spectral Information**: The OCO-2 satellite instrument measures high-resolution spectra in three spectral bands centered
around 0.76, 1.6, and 2.0 µm, referred to as the $O_2$-A, weak $CO_2$ (WCO$_2$), and strong $CO_2$ (SCO$_2$) bands, respectively (OCO-2
Science Team et al., 2019). However, only the WCO$_2$ and SCO$_2$ bands are used as inputs for current $XCO_2$ retrievals. The $O_2$-
A band is excluded as it lacks significant information needed to directly estimate $XCO_2$, based on radiative transfer principles.
Instead, the $O_2$-A band is primarily used in OCO-2's operational full-physics algorithm for rapid cloud and aerosol screening
prior to $CO_2$ retrieval (O'Dell et al., 2012), effectively excluding observational cases that potentially lead to poor retrieval
quality, thus saving substantial computational costs. Each OCO-2 spectral band is sampled by 1024 detector pixels. However,

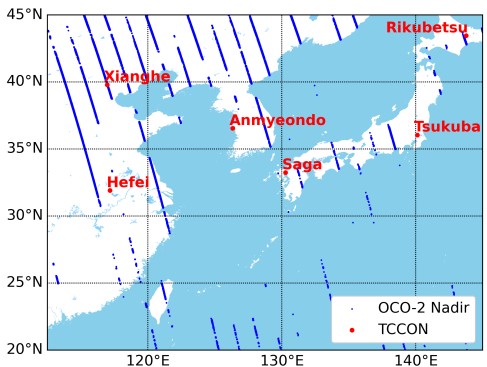

**Figure 1.** The target area for the East Asia region, distribution of observation points (from OCO-2 L2std v10r files) of OCO-2 Nadir mode
in January 2016, and the distribution of TCCON sites in this area.

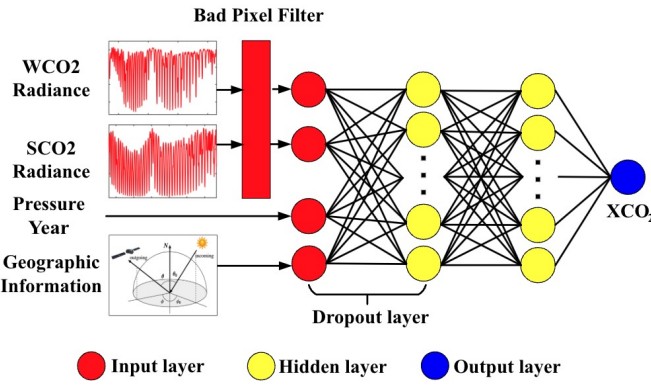

**Figure 2.** Schematic diagram of the MLP-XCO$_2$ model. The input layer includes two interpolated radiance of WCO$_2$ and SCO$_2$ band filtered through a bad pixel filter, geographical observation information, surface pressure, and the corresponding year. A dropout layer with a 0.1 dropout rate is added between the input layer and the first hidden layer.

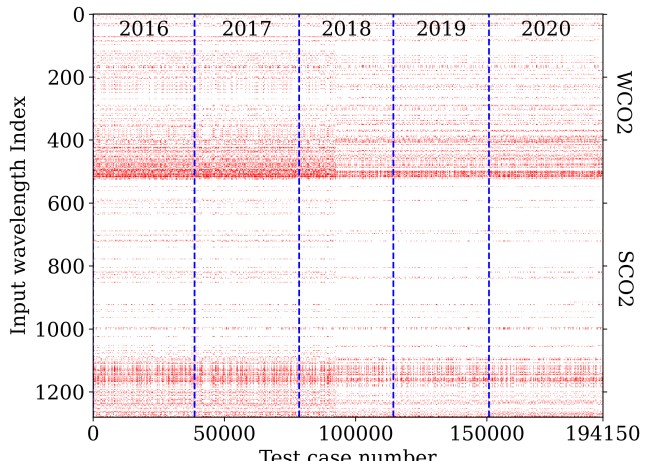

**Figure 3.** Visualization of the OCO-2 satellite data quality across interpolated wavelength grid indices. The map illustrates the bad sample list extracted from OCO-2 Level 1B files for all test cases. On the horizontal axis, sample numbers range from 0 to 194,150, while the vertical axis represents various wavelength grid indices, ranging from 0 to 1,280. Red coloration indicates problematic data pixels.

over time some detectors have degraded or become unstable in the space environment, resulting in pixels being flagged as "bad samples" in quality filters (Marchetti et al., 2019). To maximize high-quality training data, additional preprocessing is performed on the WCO$_2$ and SCO$_2$ bands. Initially, the beginning and ending spectral ranges corresponding to the most degraded detectors are removed. The remaining spectra are re-sampled into 525 and 755 wavelength points for the WCO$_2$ and SCO$_2$ bands, respectively (spectral points in wavelength are detailed in Table 2). To enhance the CO$_2$ absorption line

**Table 1.** Detailed lists of the input parameters for the MLP-XCO$_2$ model

| Input elements | Variables | Number |
|---|---|---|
| Spectral information | WCO$_2$ | 525 |
| | SCO$_2$ | 755 |
| | Bad pixel mask | 1280 |
| Geographical information | Solar zenith | 1 |
| | Relative azimuth | 1 |
| Others | Year | 1 |
| | Pressure | 1 |
| Total | | 2564 |

**Table 2.** Wavelength spacing of the input spectra

| Band | Spectral range [μm] | Spectral points [μm] |
|---|---|---|
| WCO$_2$ | 1.5990-1.6151 | $\lambda_1 = 1.5990$, $\lambda_{i+1} = \lambda_i + 10^{-4}(6.10 - 3.60\lambda_i)$, $i$ = 1-524 |
| SCO$_2$ | 2.0478-2.0779 | $\lambda_1 = 2.0478$, $\lambda_{i+1} = \lambda_i + 10^{-4}(7.58 - 3.48\lambda_i)$, $i$ = 1-754 |

information, each input spectrum is normalized by dividing the mean radiance within a nearby spectrally transparent window lacking absorption features (1.6056-1.6059 μm using 10 points for WCO$_2$; 2.0602-2.0607 μm using 15 points for SCO$_2$). Additionally, as shown in Fig. 3, it's noticed that some isolated pixels within the main CO$_2$ absorption bands still consistently exhibit poor radiance quality. To address this issue, a 'bad sample filter' has been implemented, which utilizes a binary record from the OCO-2 L1B database (0 indicates spectra derived from good quality pixels, and 1 indicates pixels with defects or derived from poor quality interpolations). The settings of this filter are determined solely by the historic records and the version of the bad pixel map, ensuring refined data quality and consistency across different versions of the map. To further address bad samples resulting from natural degradation, we've implemented a dropout layer between the initial and the first intermediate MLP layer, thus enhancing the model's generalizability with the remaining spectral inputs.

**Geographical Information**: The model is designed to accept two key observation geometry angles that are determined by the relative positions of the Sun, satellite, and ground observation point. These include solar zenith angle and relative azimuth angle. The solar zenith angle (SZA) features prominently as a cosine term in the radiative transfer equation that defines the atmospheric radiative processes. Thus, SZA is pre-converted to its cosine form for model input. The relative azimuth angle is a comprehensive angle that jointly combines the solar azimuth angle and the satellite azimuth angle. It is important to emphasize that the satellite zenith angle is not utilized in this study. Our current research is based on the Nadir mode of the OCO-2 satellite observation. In the Nadir observing mode, the satellite zenith angle is assumed to be nearly perpendicular to the Earth's surface, theoretically approaching zero degrees.

**Other parameters**: In addition to the primary inputs, two other parameters play critical roles in the MLP-XCO$_2$ model: the surface pressure, and the corresponding year (e.g., 2016 or 2017, etc.). In traditional retrieval algorithms based on iterative optimization, accurate surface pressure and a reliable prior CO$_2$ profile are crucial. The importance of this has been highlighted by the averaging kernel utilized in the OCO-2 retrieval algorithm (Braverman et al., 2015), which indicates a higher sensitivity near the surface compared to the stratosphere. To prevent the retrieval of unrealistic CO$_2$ profiles, the prior covariance matrix imposes significantly stricter constraints in the stratosphere than in the troposphere (O'Dell et al., 2012). In cases where the prior CO$_2$ profile is inadequate, it can lead to poor results, with minimal or even opposite updates in the stratospheric CO$_2$ profile during the inversion process (Iwasaki et al., 2019). Additionally, in order to achieve the best agreement between observed and estimated spectra, the retrieval process may inaccurately estimate tropospheric CO$_2$ profiles. To tackle these challenges, our investigation suggests that incorporating additional information such as the "year" can offer valuable context for XCO$_2$ retrieval. This conservative approach provides a simple means to enhance prior CO$_2$ information without directly specifying XCO$_2$ prior values.

## 3 Satellite product data based machine learning model

In this section, we first developed the MLP-XCO$_2$ model using the OCO-2 v10r product dataset. The primary goal was to optimize the hyperparameters of the MLP-XCO$_2$ network. On one hand, we aimed to confirm whether the "slow bias", as shown in Bacour et al. (2023) is a universal issue across machine learning models with similar architectures. On the other hand, by fixing the hyperparameters of the MLP-XCO$_2$ network structure, we sought to develop a comparable model using simulated data in later sections. In theory, MLP models using identical hyperparameters should possess the same fitting and generalization abilities. By first presenting results from a model trained solely on satellite product data, we can demonstrate the limitations of these satellite data-based models. This then motivates the development of new machine learning strategies to overcome these limitations, as discussed in later sections.

Following the target areas and data screening methods discussed previously, observational data and lists of bad pixels from the OCO-2 v10r L1B database. Additionally, retrieved surface pressure and XCO$_2$ data were obtained from the L2std database. Specifically, we obtained data from March, June, September and December spanning the years 2016 to 2020. This timeframe was chosen to provide a comprehensive training and testing set for our analysis. In total, the dataset encompassed 194150 samples collected over this five-year period. The year-wise distribution of the samples is as follows: 38626 samples from 2016, 39850 from 2017, 35945 from 2018, 36452 from 2019, and 43277 from 2020.

After completing the data collection, we proceeded to construct the MLP-XCO$_2$ model. To balance model complexity and performance, the MLP-XCO$_2$ architecture (Fig. 2) comprises five hidden layers, with 1000, 500, 300, 100 and 20 nodes, respectively. All hidden layers also use ReLU activation functions. The output layer contains a single node to predict XCO$_2$ values, with a linear activation function. Upon developing the MLP-XCO$_2$ model architecture, in this section, we independently trained two versions of the MLP-XCO$_2$ model, each based on the aforementioned model structure but with different training and testing datasets.

**Historical Data Training**: The first MLP-XCO$_2$ model based on OCO-2 product data was trained using historical XCO$_2$ collected from 2016 to 2018. The test set for this model comprised product data from the years 2019 and 2020. This setup allows us to assess the model's predictive performance using a straightforward historical data approach.

**Skipped-Year Training**: The second version of the model was trained using data from the years 2016, 2018, and 2020. The test set for this MLP-XCO$_2$ model included the skipped years, 2017 and 2019. This unique approach enables a clearer and 205 more direct comparison of the potential limitations of relying solely on historical data for future predictions.

Figure 4 presents the results for the two trained MLP-XCO$_2$ models on their respective 10% out-of-sample testing datasets. The first subplot illustrates the predictions of the historical data training model from the 2016-2018 data, and the second subplot shows similar predictions for the skipped-year training model. Both models achieve high accuracy on these testing datasets, with a Root Mean Square Error (RMSE) close to 1 ppm and an R-squared score (R$^2$) larger than 0.9. These results demonstrate 210 the robust interpolation capabilities of both models within their respective training periods, indicating their effectiveness in handling known observed scenarios.

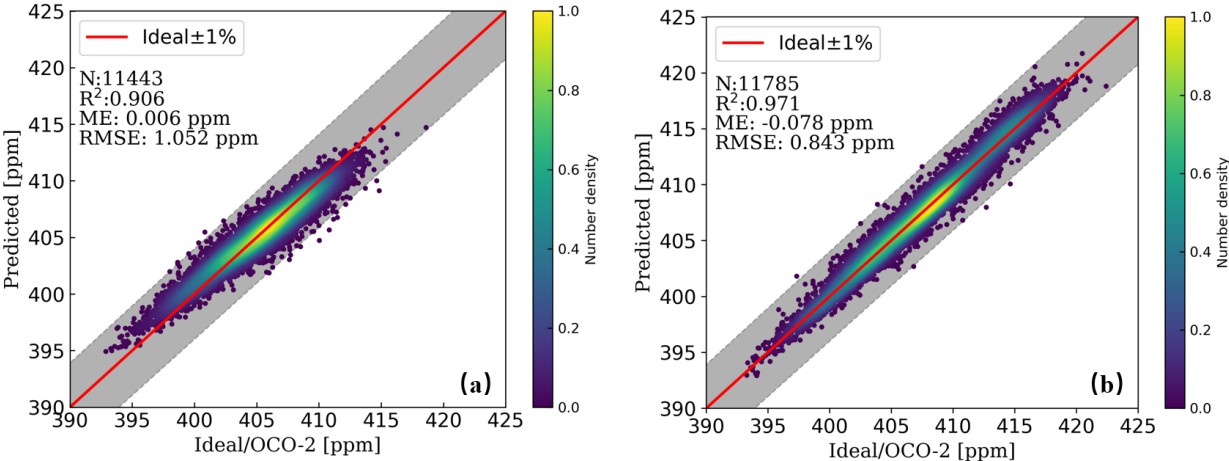

**Figure 4.** Comparison of 10% out-of-sample XCO$_2$ testing cases predicted by the MLP-XCO$_2$ model versus results retrieved by OCO-2 v10r product. Panel (a) is for the historical data training model, while (b) is for the skipped-year training model. The solid red lines in the figure correspond to the perfect agreement, where shadow areas around the solid red lines represent $\pm 1\%$ of XCO$_2$ deviations.

Figure 5 evaluates the generalization capabilities of each MLP-XCO$_2$ model on testing sets comprising years not included in their respective training datasets. These test sets represent periods outside the range of years used for training. Here, we solely observed a noticeable positive bias in the predictions from the historical data training model. In contrast, the skipped-year 215 training model did not exhibit this bias. Performance remains highly accurate on these out-of-range points, further validating the model's robustness for XCO$_2$ prediction within skipped years.

Globally, the average XCO$_2$ in the atmosphere shows a stable annual increase, with an observed rise of approximately 2-3 ppm per year. However, despite the inclusion of the corresponding year in the input layer as a high-correlation parameter,

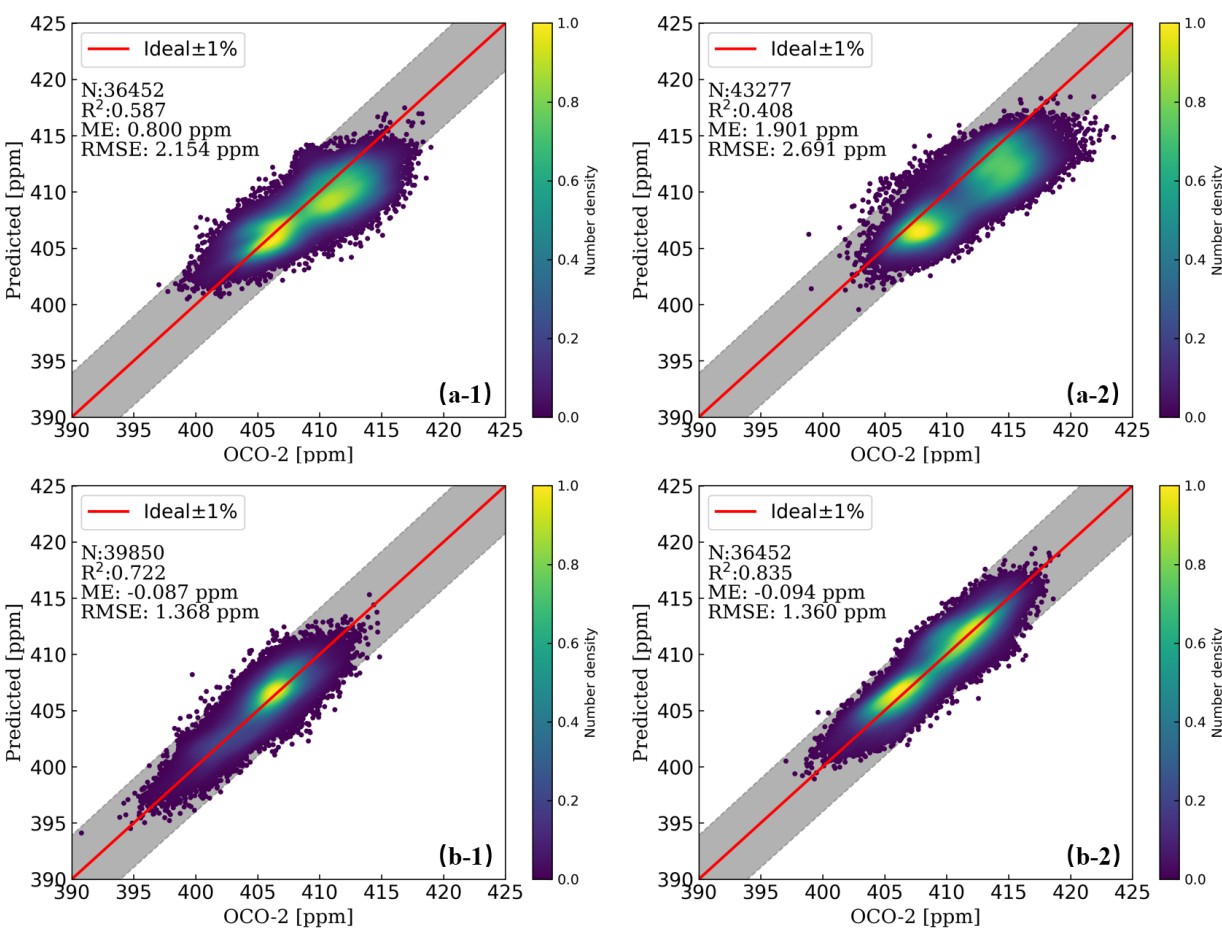

**Figure 5.** Comparison of XCO$_2$ results predicted by the MLP-XCO$_2$ model versus results retrieved by OCO-2 v10r product on test sets consisting of years not included in the training periods. Panels (a-1) and (a-2) are for the historical data training model on the 2019 and 2020 test sets, respectively. Panels (b-1) and (b-2) are for the skipped-year training model on the 2017 and 2019 test sets, respectively. The solid red lines in the figure correspond to the perfect agreement, where shadow areas around the solid red lines represent ±1% of XCO$_2$ deviations.

there remains a limitation in capturing the atmospheric CO$_2$'s potential rising trend. This highlights the limitations of models trained solely on historical satellite data, motivating the development of new techniques to incorporate external information about temporal CO$_2$ dynamics.

## 4 Simulation data based machine learning model

In the previous section, the MLP-XCO$_2$ model showed excellent interpolation within the training data range but exhibited bias when predicting outside this period. To eliminate this bias, we propose using an accurate forward model to simulate

representative training data that covers future atmospheric conditions. If we can pre-generate atmospheric profiles that capture possible future states, and simulate their corresponding spectral radiance using an accurate forward model, the MLP-XCO$_2$ model can pre-learn future satellite observations. This could prevent the incremental annual bias and enable accurate XCO$_2$ prediction. The effectiveness of this approach depends on the forward model accuracy and representativeness of the simulated atmospheres (Zhao et al., 2022).

It is therefore critical to select an appropriate radiative transfer forward model with proven reliability in simulating spectral radiance under varying atmospheric conditions. The model must precisely capture the relationship between trace gas concentrations, meteorological states, and resulting spectral signatures. With accurate simulations, the machine learning model can generalize robustly to future atmospheric scenarios. The representative training data should span the expected range of atmospheric variability in XCO$_2$ and interfering species like water vapor. A broad sampling of the state space is key for the model

to learn a robust mapping to XCO$_2$ across multiple atmospheric regimes. The following sections describe our approach for accurate spectral radiative transfer simulations and possible (realistic) atmospheric profile generations.

## 4.1 Forward model

In this study, we developed a forward radiative transfer calculation model using the ReFRACtor (Reusable Framework for Retrieval of Atmospheric Composition) software (McDuffie et al., 2018). ReFRACtor is an extensible framework for multi-

240 instrument atmospheric radiative transfer and retrieval, originally derived from the operational OCO-2 retrieval program. Although ReFRACtor contains both radiative transfer and retrieval capabilities, we only utilized the radiative transfer component. Specifically, we configured ReFRACtor to simulate top-of-atmosphere radiance spectra that would be observed by OCO-2. These simulated observations were then used to generate a large training dataset for our machine learning model, MLP-XCO$_2$.

The OCO-2 satellite primarily observes the radiative spectra in the short-wave infrared (SWIR) band. Over the range of

245 SWIR, the impact of thermal emission can be ignored when simulating the spectra (Crisp et al., 2021). To simulate OCO-2's observed spectra in the WCO$_2$ band around 1.6 μm and the SCO$_2$ band around 2.06 μm, the ReFRACtor model numerically solves the Eq. (1) of the radiative transfer equation (RTE) (Modest and Mazumder, 2021):

$$\mu \frac{dI(\tau,\mu,\phi)}{d\tau} = -I(\tau,\mu,\phi) + J(\tau,\mu,\phi) \tag{1}$$

where $I_\eta$ is the observed spectra, $\mu$ is the cosine of the observation zenith angle (e.g., $\mu = \cos\theta$), $\tau$ is the vertical optical depth

which can be column-integrated from the molecular absorption coefficients and optical path, $\phi$ is the azimuthal angle relative to the observation point for the satellite and the sun, and $J$ represents the scattering components and inhomogeneous source term, describing both single scattering and multiple scattering contributions. The term $J$ in RTE can be expressed as Eq. (2):

$$J(\tau,\mu,\phi) = \frac{\omega}{4\pi} \int\limits_{-1}^{1} \int\limits_{0}^{2\pi} P(\tau,\mu,\phi;\mu',\phi') I(\tau,\mu',\phi') d\mu' d\phi' + \frac{\omega}{4\pi} P(\tau,\mu,\phi;\mu',\phi') I_0 \exp(-\tau/\mu_0) \tag{2}$$

where $\omega$ is the single scattering albedo, $P$ is the scattering phase function, $\mu'$ and $\phi'$ are the cosine and azimuth angle of the

255 incident direction angle in each direction, $\mu_0$ is the cosine of the solar zenith, and $I_0$ is the solar intensity in the top of the atmosphere.

The ReFRACtor model uses a hybrid model to solve RTE. Specifically, the radiative transfer software LIDORT (Spurr, 2008) is applied for the scalar and Jacobian computation. Concurrently, the two-order scattering model (Natraj and Spurr, 2007) is utilized for the additional radiance correction. Within this framework, the ReFRACtor model comprehensively considers five types of scatter particles for each sounding: two types of clouds, two types of tropospheric aerosols, and one type of stratospheric aerosol. The single scattering optical properties for each cloud and aerosol particle, including cross-section, single scattering albedo, and scattering phase matrix, have been pre-computed and tabulated for the forward calculations. Furthermore, the model determines surface reflectance as a quadratic spectral albedo for each band which is derived from the bidirectional reflectance distribution function (BRDF).

An essential step for developing the forward calculation model is referencing the pre-computed look-up table of $H_2O$ and $CO_2$ to obtain the required spectral absorption coefficients. In this study, the ABSCO v5.1 database (Absorption Coefficient Table (Payne et al., 2020)) was applied for this purpose. Additionally, we identified and corrected an overestimation of the solar continuum in ReFRACtor compared to the OCO-2 Level 2 algorithm (Crisp et al., 2021). Without this correction, there would have been approximately 3 % overestimation in the 1.6 μm band and 6.5 % in the 2.06 μm band. By reducing the solar continuum, our forward model aligned better with the OCO-2 spectral measurements. These configurations of the absorption coefficients and solar continuum were essential to accurately simulate OCO-2 spectra for generating training data across a variety of observing conditions.

To assess the performance of the forward model, we selected four distinct global locations in the year of 2017. The goal was to replicate the OCO-2 observed spectra for both the $WCO_2$ 1.6 μm absorption band and the $SCO_2$ 2.06 μm absorption band at the four locations. By accessing the OCO-2 L2std database, we acquired atmospheric conditions and pertinent geographical data (including spectral albedo, surface pressure, and observation angles) specific to these chosen locations. The outcomes of our simulations for these four locations are visually depicted in Fig. 6 and Fig. 7, respectively for the two bands, with accompanying residual plots displayed in the lower panels. It is worth noting that the simulated results exhibit a high level of agreement with the observed OCO-2 spectra; the relative error remains under 1%, underlining the robustness of the established forward model. The remarkable agreement between the observed and simulated spectra indicates the excellent performance of the forward radiative transfer model. This performance is particularly evident in accurately replicating the satellite observations from OCO-2. As a result, this forward model serves as a reliable tool for the development of machine learning models trained using simulated spectral data.

## 4.2 Training data generation

To optimize the training of the MLP-$XCO_2$ model, it is essential that the input training vectors cover a wide range of realistic variations. Although the idea of randomizing all input parameters to enhance diversity might appear attractive, simulating satellite spectra involves managing a multitude of interdependent variables. In addition to the $CO_2$ vertical profile, factors such as surface pressure, temperature profile, water vapor, aerosols, and observation geometry must be accurately represented. Randomizing all of these parameters would require an impractical amount of data and could result in combinations that have no real-world relevance. For example, the four viewing angles determined by the sun, observation point, and the OCO-2

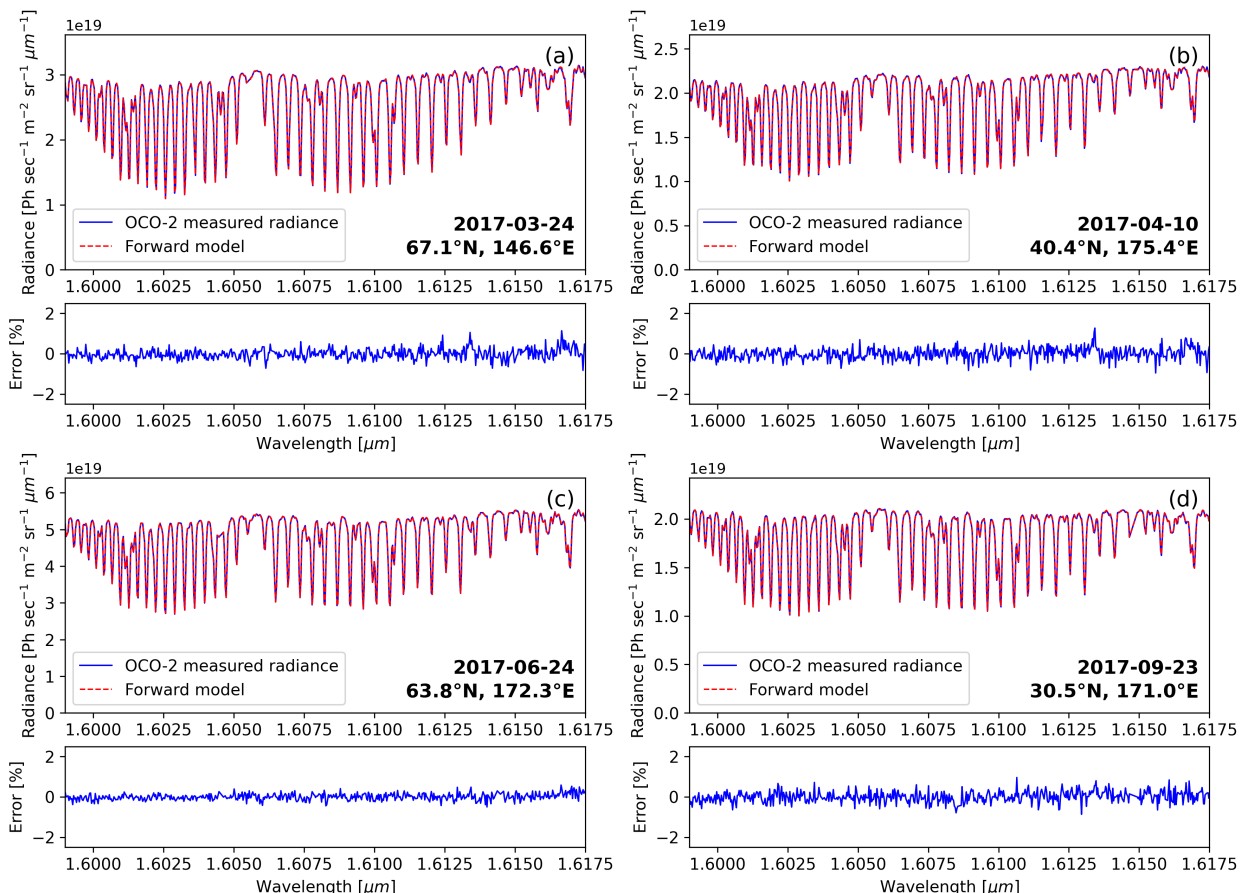

**Figure 6.** Comparisons of the OCO-2 observed spectra with the simulated ones from the modified ReFRACtor forward calculation model in WCO$_2$ band. The lower panel shows the relative error between the spectrum observed by the OCO-2 satellite and that simulated by the forward calculation model. Subplots (a)-(d) correspond to test samples from four different regions. The input vectors for the ReFRACtor model were derived from OCO-2 L2std retrieved results.

satellite have fixed combinations during the satellite's regular operation. Therefore, randomly selecting angle combinations lacks practical significance. To ensure that the training data covers valid variations, we conducted an analysis of historical OCO-2 retrievals. This analysis revealed consistent seasonal patterns and year-to-year trends in most parameters. This supports the idea of selecting representative samples from statistical distributions rather than relying on complete randomization. By
295 carefully considering the relationships between parameters and the realities of satellite observations, we can create a reasonably sized training dataset that effectively captures the range of expected predictions.

The generation of the vertical CO$_2$ profile is especially critical among all input parameters. This dataset theoretically determines the generalization domain of the MLP-XCO$_2$ model. In the forward model based on the ReFRACtor model, the atmospheric CO$_2$ profile is segmented into 20 sub-layers by pressure. By statistically analyzing the OCO-2 retrieved CO$_2$

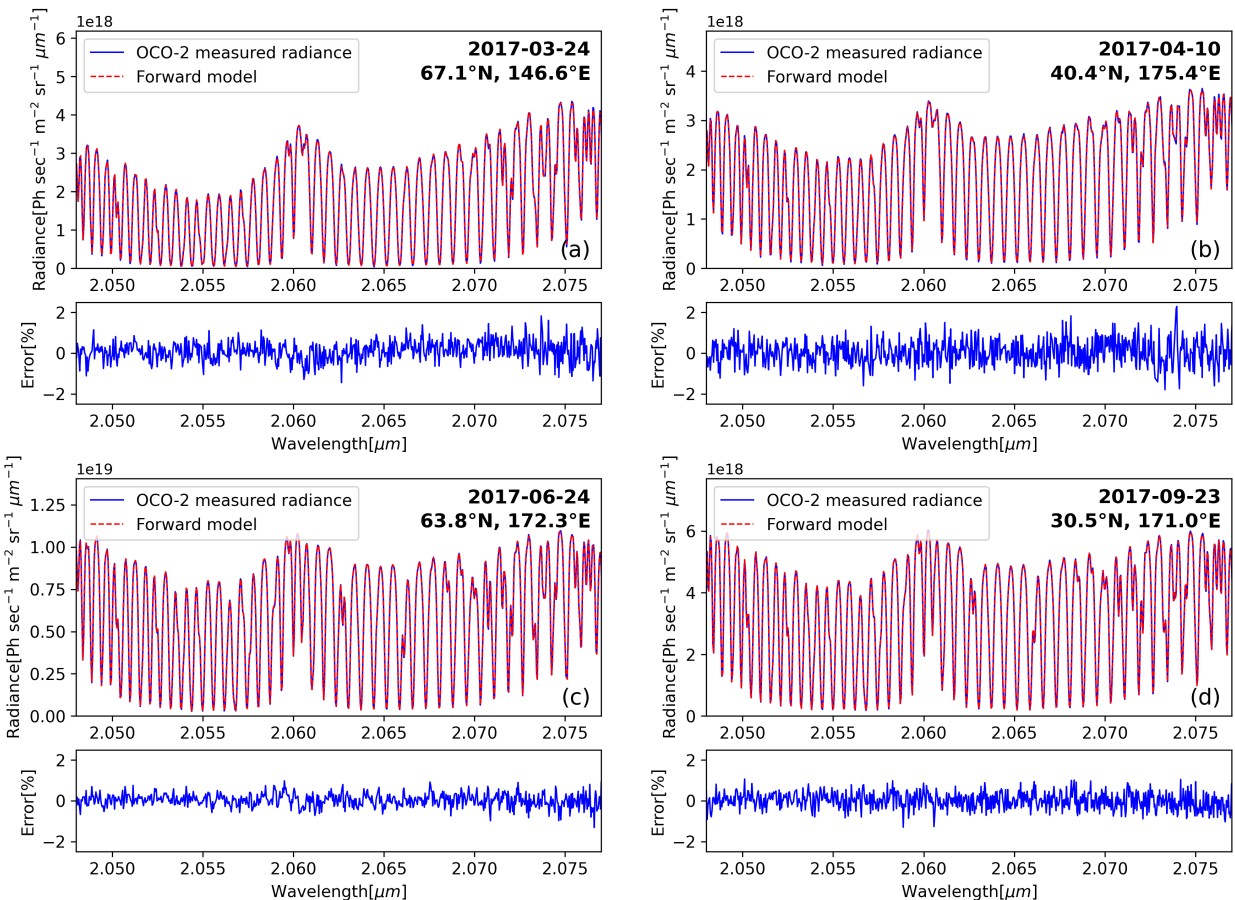

**Figure 7.** Comparisons of the OCO-2 observed spectra with the simulated ones from the corrected ReFRACtor forward calculation model in $SCO_2$ band. The lower panel shows the relative error between the spectrum observed by the OCO-2 satellite and that simulated by the forward calculation model. Subplots (a)-(d) correspond to test samples from four different regions. The input vectors for the ReFRACtor model were derived from OCO-2 L2std retrieved results.

profiles in the target East Asia area from 2016-2018, the box plots for atmospheric $CO_2$ concentration in each sub-layer are shown in Fig. 8(a), and the historic $XCO_2$ results from the OCO-2 product data are showing in Fig. 8(b). From the upper atmosphere down to the ground surface, the variability of $CO_2$ concentrations gradually increases. This challenges the ability for the standardization of atmospheric $CO_2$ profiles, particularly closer to the Earth's surface. Fortunately, a consistent year-on-year rise in $CO_2$ concentrations in each sub-layer has been observed over time. Consequently, in our research, we have proposed a method for generating subsequent $CO_2$ atmospheric profiles. We incrementally increase the $CO_2$ concentration by 2.5 ppm annually, starting from the 2016 OCO-2 retrieved $CO_2$ vertical profile. This approach ensures that we encompass a range of plausible atmospheric $CO_2$ distributions with realistic shapes, enabling the generation of simulated spectra for the designated training years.

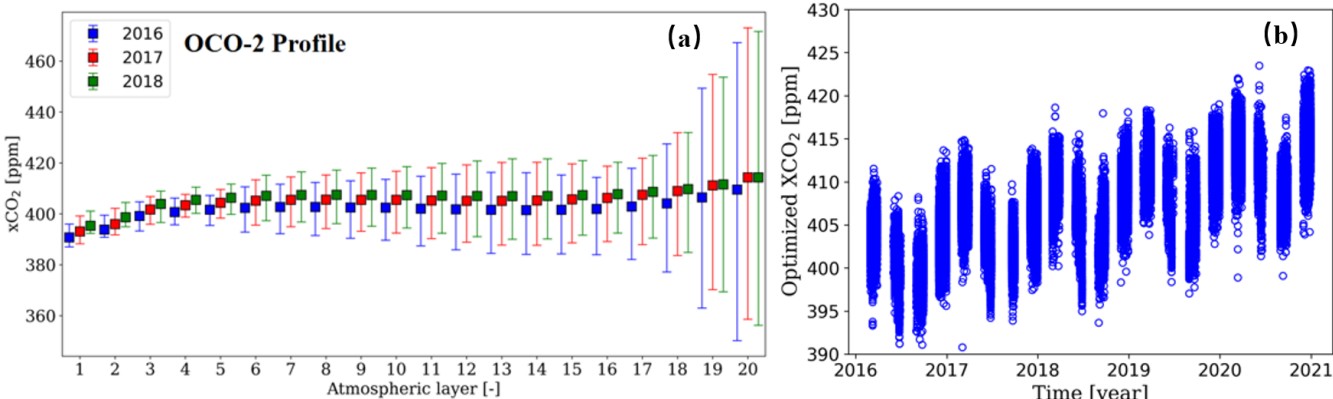

**Figure 8.** Panel (a) is the boxplot of the vertical distribution of $CO_2$ profiles (from OCO-2 L2std files) retrieved by the OCO-2 satellite over East Asia in Nadir mode from 2016 to 2018. The horizontal axis represents the atmospheric layers from layer 1 (top of the atmosphere) to layer 20 (near-surface). The upper and lower bounds of each box show the maximum and minimum $CO_2$ concentrations recorded within that layer for each year. Panel (b) is the scatter plot of historic $XCO_2$ results retrieved by the OCO-2 inversion program (from L2std files)

.

In addition to the $CO_2$ profile, Fig. 9 illustrates the year-to-year trends of various observed parameters essential for the forward calculation model in the East Asian region. These parameters, although they display seasonal variations, consistently exhibit annually cyclic patterns. Given that the OCO-2 satellite conducts global observations in cycles of approximately half a month (15-16 days), this study employed observation parameters and priori data for atmospheric profiles, except for $CO_2$, from the year 2016 as a reference. These reference data were repetitively utilized for generating simulations in subsequent years. Regarding the quadratic spectral albedo, the constant term in the training data samples is uniformly set to 1 (to be normalized before being processed by the neural network). The slope and the quadratic coefficient are stochastically sampled within the range of values corresponding to the retrieval results based on the OCO-2 L2 products.

Based on 60,000 uniformly sampled observation data points exclusively from the OCO-2 satellite throughout 2016, we randomly separated it into six sets of 10,000 data points each. Each set represents $CO_2$ profiles from 2016 to the end of 2021, with a yearly increase of 2.5 ppm added to the original data, reflecting projected future profiles. The forward model was used to generate the corresponding simulated spectra for each set. These simulated samples serve as the foundational dataset for training the new MLP-$XCO_2$ machine learning model. It's important to note that this new model relies solely on the data recorded by the OCO-2 satellite in 2016, as its reference. However, it is essential to acknowledge that real-world observations by the OCO-2 satellite involve parameters that are not predetermined in future simulations, such as the Empirical Orthogonal Functions (EOF) parameters, signal-to-noise ratio (SNR), bad sample lists, and the degradation of grating pixels. Therefore, our new model is trained not only on the 60,000 simulated datasets but also on the 2016 historical data. According to the data selection criteria outlined in Section 2.1, we identified a total of 38,626 sets of historical data in 2016, comprising spectral measurements from OCO-2 and the corresponding $XCO_2$ products. These historical experimental datasets are integrated with

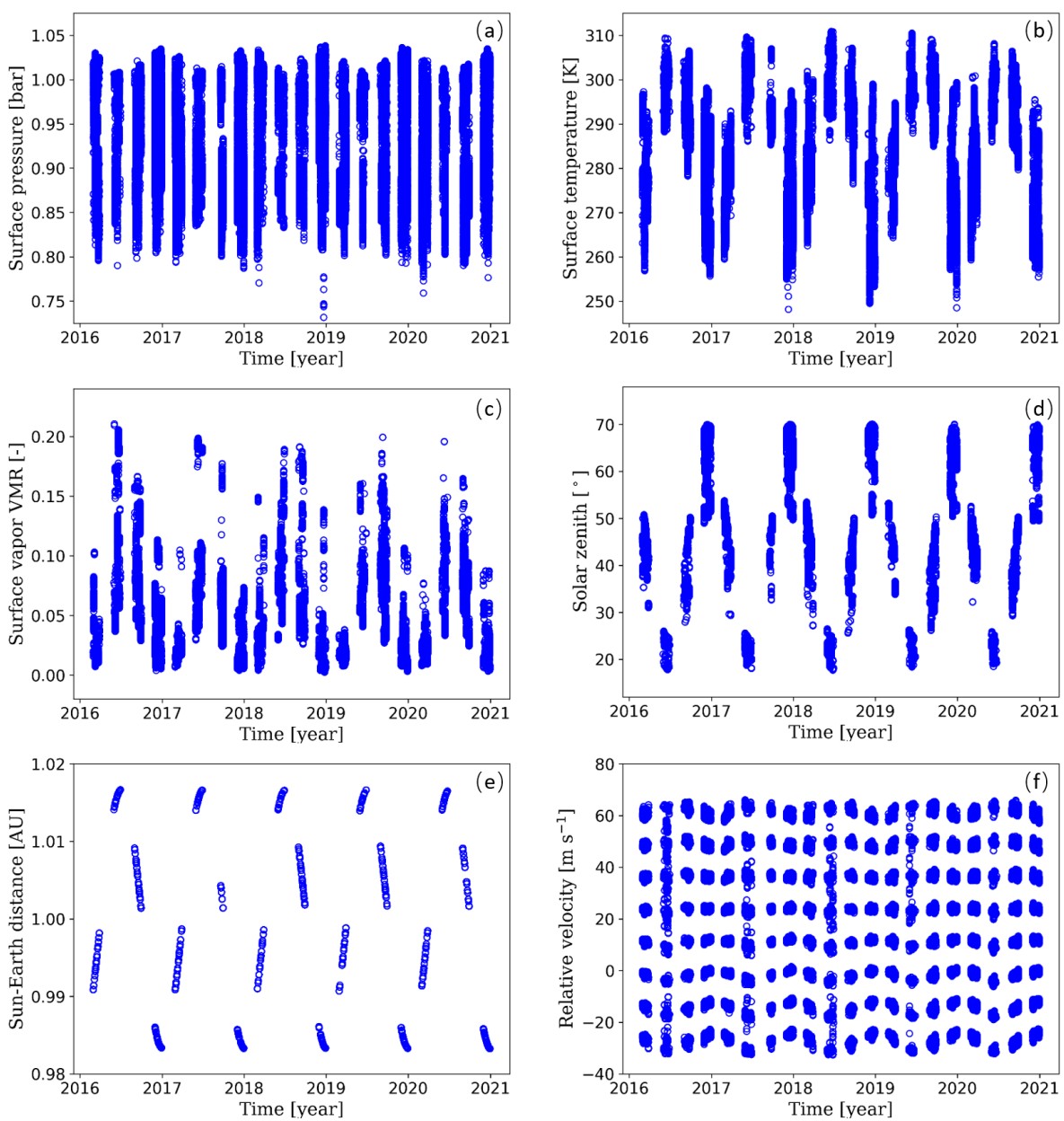

**Figure 9.** Scatter plots of atmospheric parameters required for forward calculation models (excluding $CO_2$ profiles) from 2016 to 2020, sourced from the OCO-2 L2 product. Panel (a) is the surface pressure, (b) is the surface temperature, (c) is the near-surface water vapor concentration, (d) is the solar zenith, (e) is the Sun-Earth distance, and (f) is the Earth-satellite relative velocity.

the simulated data, enriching the training datasets. This dual combination and data augmentation techniques ensure that the model is well-equipped to handle both potential future atmospheric conditions and the current realities of instrument and

330 spectral measurement capabilities. By doing so, we provide a more comprehensive training strategy that captures both the anticipated future scenarios to accurately and efficiently perform $XCO_2$ retrieval for the "future" years from 2017 to the end of 2020.

## 5 Results and discussions

### 5.1 Comparison with the OCO-2 satellite product data

To evaluate the retrieval capability of the MLP-$XCO_2$ model trained on a combined dataset of simulated data and historical 2016 OCO-2 satellite data, the neural network architecture and hyperparameters were intentionally kept identical to the previous model trained solely on actual OCO-2 satellite product data. Keeping these factors constant isolates the training data source as the only major difference between the models. This enables a direct, apples-to-apples comparison of how the training data affects model performance.

Figure 10 (a) shows the retrieval results on 10% out-of-sample testing data that was excluded from model training. Setting aside this test subset is a standard technique for evaluating model performance on new examples. The accurate predictions of the MLP-$XCO_2$ model on the test data suggest the model has learned generalizable patterns not overfit to the training data. Figure 10 (b) shows the comparison of the retrieval results of the MLP-$XCO_2$ model on real OCO-2 satellite spectral observations in 2016. Figure 11 displays $XCO_2$ predictions from 2017 to 2020 using test data consistent with Fig. 4 and Fig. 5.

As the simulated training data was generated based on 2016 OCO-2 measurements, testing on 2017-2020 data evaluates the model's ability to make predictions beyond the time frame of the training data. The scatter plots demonstrate the MLP-$XCO_2$ model trained on simulated data can accurately and stably predict the annual $XCO_2$ growth trend, maintaining RMSE less than 1.8 ppm (0.45%). Compared to models trained relying solely on historical satellite product data, the key advantage is the ability to make reasonable forecasts of future atmospheric $XCO_2$ levels.

Table 3 offers a detailed spatio-temporal comparison of the results presented in Figure 11, enhancing our understanding of the MLP-$XCO_2$ model's performance across distinct subregions within East Asia. This table specifically focuses on a finer spatial segmentation within the broad East Asian longitude and latitude range, dividing it into four subregions. These are defined based on the geographical demarcation of 35°N and 130°E, categorized as Northeast (NE), Northwest (NW), Southeast (SE), and Southwest (SW) regions, respectively. The results demonstrate that, regardless of the distribution of sample sizes across

these subregions and their varied topographical characteristics (land or ocean), the model maintains a consistent and stable performance in each subregion. Furthermore, the error metrics for these individual subregions align closely with the overall regional errors, indicating a uniformity in the model's predictive accuracy and reliability across different spatial scales within East Asia.

Considering these results, by generating possible realistic future prior information for the atmospheric conditions and us-

360 ing an accurate forward model to simulate the corresponding spectra, the approach avoids inherent biases when extrapolating beyond the distribution of the training data. Rather than simply extending trends, the model is constrained by fundamental physical relationships to interpolate within realistic bounds. This transforms the prediction task into a well-posed interpolation

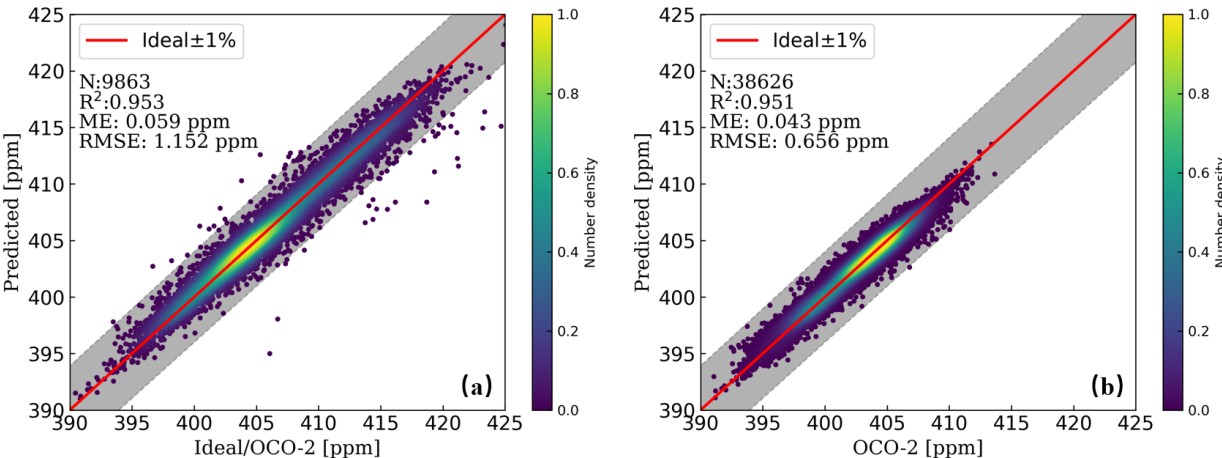

**Figure 10.** The comparison of $XCO_2$ results predicted by the MLP-$XCO_2$ model from 10% test data (not involved in training). Panel (a) shows the predicted $XCO_2$ values for the test data that are derived from the simulated dataset, and panel (b) shows the test data that are derived from OCO-2 2016 L2 $XCO_2$ data.

**Table 3.** Spatio-temporal comparison of $XCO_2$ predicted by MLP-$XCO_2$ model results versus results retrieved by OCO-2 across four subregions. These subregions are delineated based on the geographical demarcation of $35°N$ and $130°E$ as: Northeast (NE), Northwest (NW), Southeast (SE), and Southwest (SW) regions, respectively.

| Year | Full [Number / ME / RMSE] | NE | NW | SE | SW |
|------|---------------------------|-----|-----|-----|-----|
| 2016 | 38626 / 0.043 / 0.656 | 2714 / -0.037 / 0.600 | 26111 / 0.053 / 0.664 | 451 / 0.010 / 0.583 | 9350 / 0.041 / 0.651 |
| 2017 | 39850 / 0.358 / 1.563 | 1235 / 0.752 / 1.494 | 30774 / 0.392 / 1.594 | 244 / 0.701 / 1.537 | 7597 / 0.147 / 1.443 |
| 2018 | 35945 / 0.114 / 1.497 | 1854 / -0.073 / 1.356 | 27288 / 0.145 / 1.483 | 745 / 0.525 / 1.398 | 6058 / -0.015 / 1.609 |
| 2019 | 36452 / -0.242 / 1.732 | 1777 / -0.642 / 1.552 | 26082 / -0.427 / 1.863 | 304 / 0.345 / 1.432 | 8289 / 0.405 / 1.296 |
| 2020 | 43277 / -0.268 / 1.679 | 1841 / 0.111 / 1.586 | 34971 / -0.461 / 1.696 | 270 / -0.413 / 1.917 | 6195 / 0.712 / 1.595 |

problem versus an unconstrained extrapolation. The simulated data provides a physical regularization that makes the model's outputs to be scientifically sound. By training on synthetic data spanning potential future scenarios, the model learns robust rep-
resentations not tightly coupled to specifics of the training data time period. This enables high-fidelity inversion and prediction of $XCO_2$ even for future time periods beyond available measurements.

## 5.2   Detecting plume features from the OCO-2 observation

In a further effort to deeply analyze the ability of our MLP-$XCO_2$ model to capture key $XCO_2$ information from spectral data, we focused on plume detection at sites of potentially high emissions, such as thermal power plants, in our target regions from
2017 to 2020. Utilizing the data in the work of Li et al. (2023), we sourced test samples from multiple instances of $XCO_2$

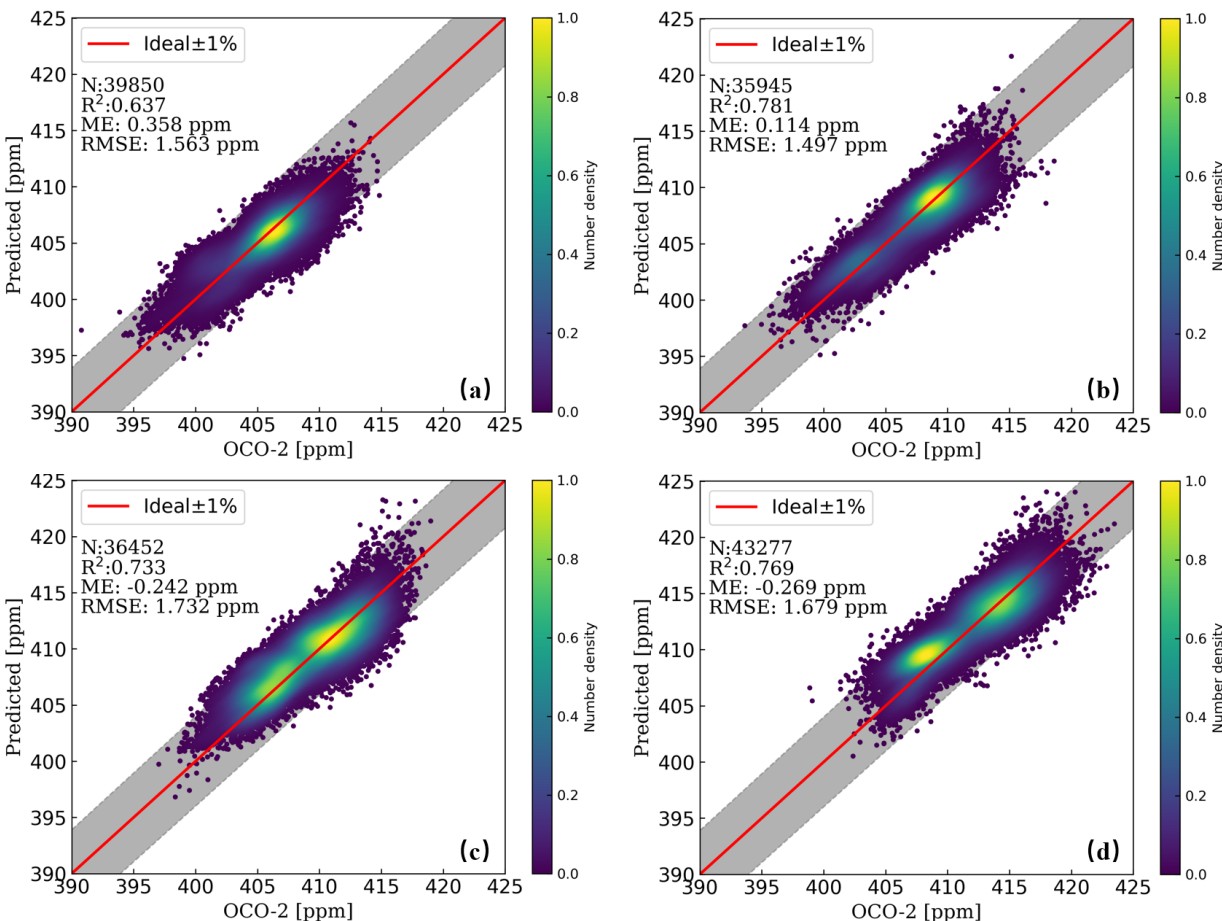

**Figure 11.** Comparison of XCO$_2$ results predicted by the MLP-XCO$_2$ model versus results retrieved by OCO-2 v10r product from 2017-2020. Panel (a), (b), (c), and (d) display the predictions of the MLP-XCO$_2$ model from 2017 to 2020, respectively.

enhancements detected by the OCO-2 satellite in Nadir mode observations. These samples were located in close proximity to known large power plants, providing an ideal scenario for assessing retrieval accuracy in detecting localized emission sources.

Figure 12 presents a geographical map that highlights XCO$_2$ predictions in the test samples from the MLP-XCO$_2$ model and compares them with results retrieved by the OCO-2 v10r product. This map clearly marks power plants with red triangles, establishing a visual link between industrial emission sources and observed points where elevated XCO$_2$ levels are detected. Figure 13 further explores this relationship by presenting a longitude-based comparison of XCO$_2$ results. This figure plots the same data points from Figure 12 against their corresponding longitude coordinates. This arrangement facilitates a direct and intuitive comparison of the trends in XCO$_2$ enhancements as captured by our model and as reported by the OCO-2 product.

In both figures above, it is visually evident that observation points near power plants show sudden increases in XCO$_2$ values, aligning with the trend from the OCO-2 v10r product. This trend is particularly pronounced when compared to points

farther away from these emission sources. Considering that these samples are nearly identical in terms of observation angles and times, such consistency is a powerful confirmation to our model's capability to retrieve genuine atmospheric $XCO_2$ from OCO-2 spectral data.

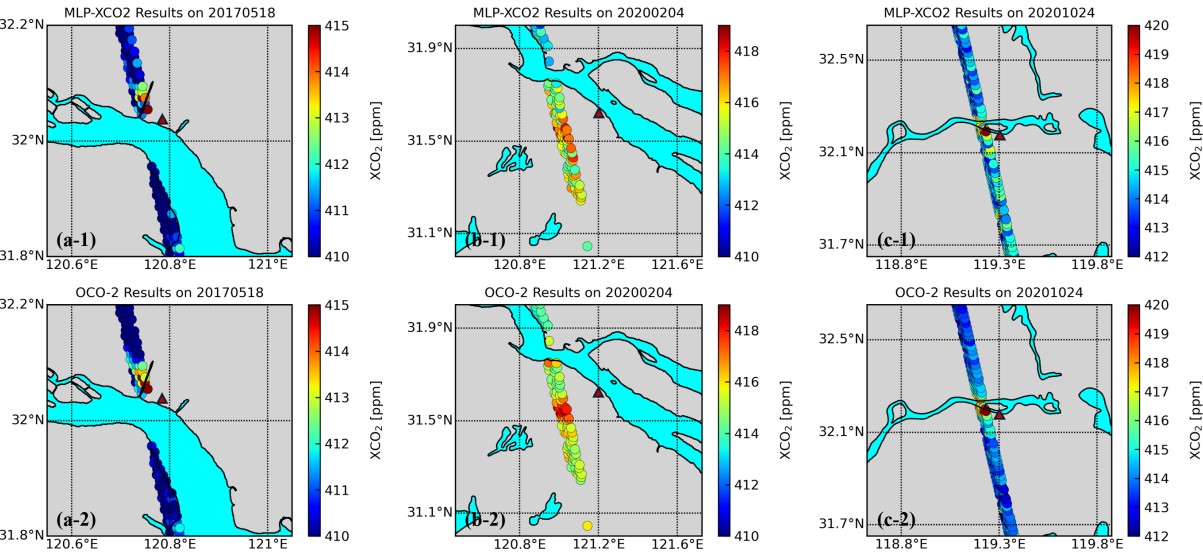

**Figure 12.** Geographical map of $XCO_2$ predictions by the MLP-$XCO_2$ model compared with OCO-2 v10r product results. The potential plume enhancements and the large power plants (marked by red triangles) were screened in Nadir mode OCO-2 observations as reported in the work of Li et al. (2023).

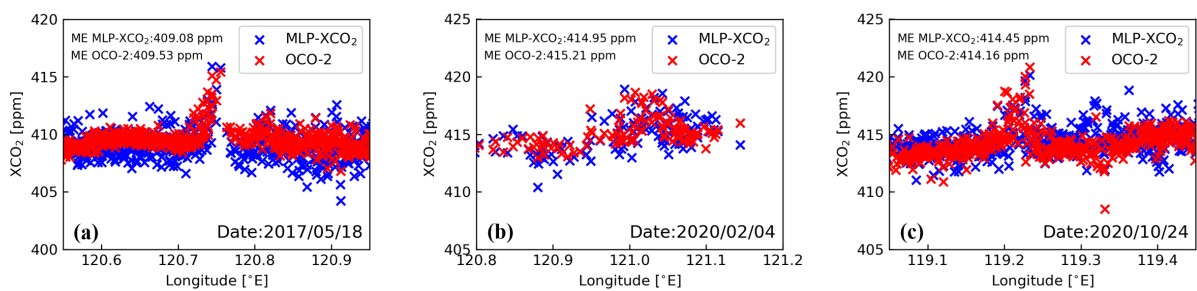

**Figure 13.** Longitude-based scatter comparison of $XCO_2$ predicted by the MLP-$XCO_2$ model versus results retrieved by OCO-2 v10r product. The potential plume enhancements were screened in Nadir mode OCO-2 observations as reported in the work of Li et al. (2023). ME represents the mean value of XCO2 within the longitude range shown in the figure.

### 5.3 Comparison with the TCCON data

A comparison of the retrieved results from the OCO-2 satellite showed that the RMSE of our developed MLP-XCO$_2$ model was around 2 ppm. In other words, our results could be worse or better than OCO-2 satellite, requiring further comparison with ground-based measurements. To further validate the accuracy of the MLP-XCO$_2$ model, we compared the XCO$_2$ retrievals from the OCO-2 v10r Nadir mode products, the MLP-XCO$_2$ model outputs, and ground-based measurements from five TC-CON sites within the study region (Fig. 1). As summarized in Table 4, spatiotemporal screening was applied to the TCCON and OCO-2 data to obtain comparable observations. The five TCCON sites included were: Tsukuba (Morino et al., 2022b), Saga (Shiomi et al., 2022), Hefei (Liu et al., 2022), Xianghe (Zhou et al., 2022) and Rikubetsu (Morino et al., 2022a). The Anmyeondo site was excluded from this analysis as the XCO$_2$ data was not updated in the TCCON GGG2020 database, and was only available until early 2018 in the GGG2014 database.

**Table 4.** Spatio-temporal screening conditions for TCCON sites and OCO-2 satellite Nadir mode observations

| TCCON site | Local time | Observed location | Sample number | Reference |
|---|---|---|---|---|
| Tsukuba | $12:48 - 12:58$ | $36.05°N \pm 0.5°, 140.12°E \pm 0.5°$ | 2078 | Morino et al. (2022b) |
| Saga | $13:30 - 13:40$ | $33.24°N \pm 0.5°, 130.29°E \pm 0.5°$ | 87 | Shiomi et al. (2022) |
| Hefei | $13:20 - 13:30$ | $31.90°N \pm 0.5°, 117.17°E \pm 0.5°$ | 984 | Liu et al. (2022) |
| Xianghe | $13:15 - 13:25$ | $39.80°N \pm 0.2°, 116.96°E \pm 0.2°$ | 2770 | Zhou et al. (2022) |
| Rikubetsu | $13:20 - 13:30$ | $43.46°N \pm 0.2°, 143.77°E \pm 0.2°$ | 723 | Morino et al. (2022a) |

Figure 14-1 presents time series comparisons of XCO$_2$ retrievals from the different TCCON sites, MLP-XCO$_2$ model, and OCO-2 Nadir observations. Figure 14-2 displays the box plots of the differences between the MLP-XCO$_2$ model results, OCO-2 products, and TCCON site data. The plots at each of the five TCCON sites demonstrate the simulated data-trained MLP-XCO$_2$ model accurately predicts XCO$_2$ from the OCO-2 spectra. The model successfully captures seasonal variations and the long-term XCO$_2$ growth trend over the 4-year study period. The reliable performance over time and across multiple TCCON sites further validates the model has learned generalizable representations of carbon cycle processes rather than overfitting to specifics of the simulated training data. By using realistic future simulations for training, the model provides robust and unbiased XCO$_2$ retrievals across a range of atmospheric conditions.

### 5.4 Retrieval efficiency

In this study, the ReFRACtor forward model required 12.16 seconds per simulation case (two absorption bands) using an AMD Ryzen-7 5800X computer. The OCO-2 retrieval based on Bayesian optimization typically needs over three iterations to converge, indicating at least 36.48 seconds per retrieval. In contrast, the MLP-XCO$_2$ model demonstrated remarkable efficiency on the same hardware. It required just 1.14 seconds total to retrieve XCO$_2$ from 6642 OCO-2 test spectra across all five TCCON sites, averaging 0.17 milliseconds per sample with GPU RTX 3080Ti. This rapid inversion drastically reduces processing times

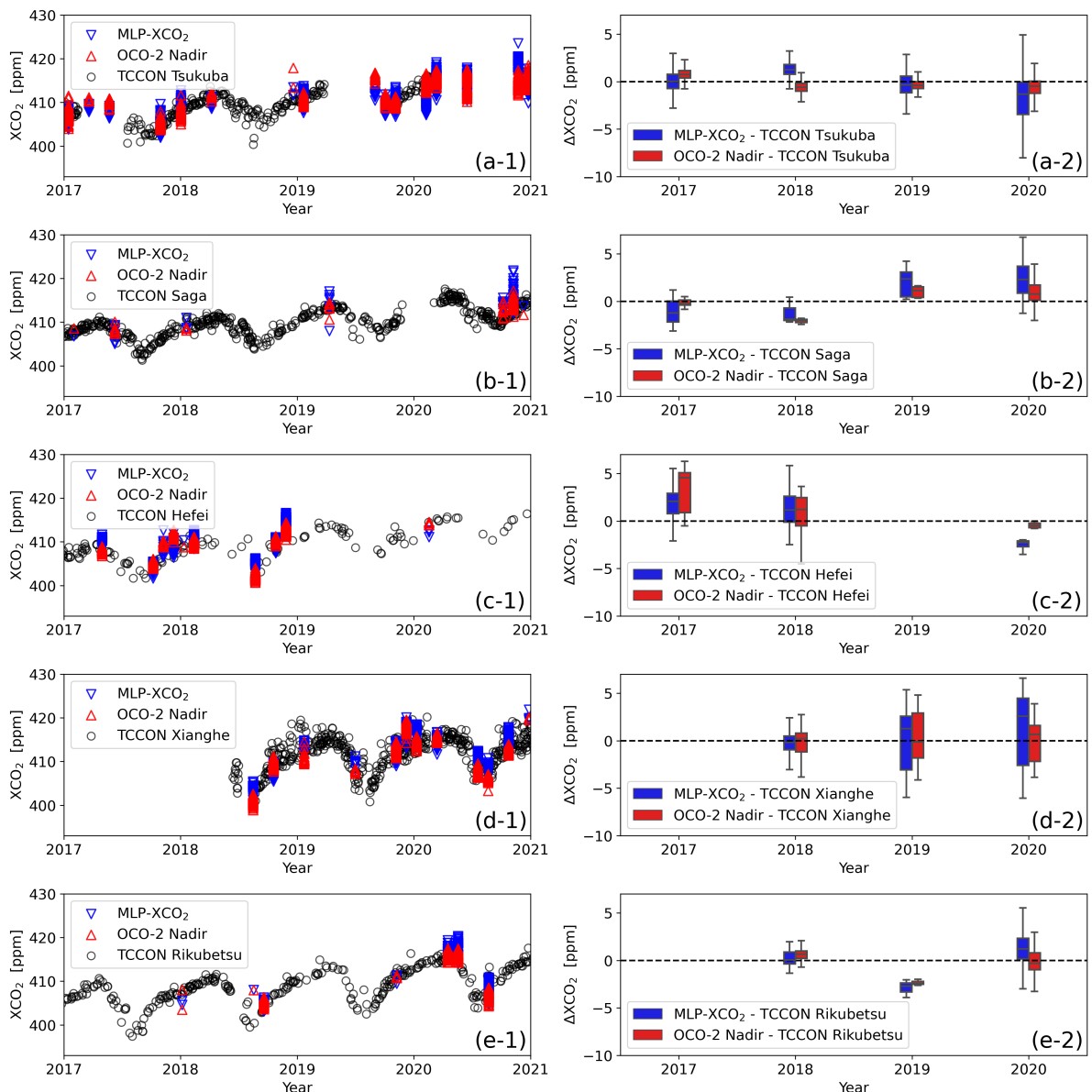

**Figure 14.** Comparisons of $XCO_2$ results from 2017 to 2020 across five TCCON sites. Panel (a-1)-(e-1) show the time series comparisons of $XCO_2$ retrievals from the different TCCON sites, MLP-$XCO_2$ model, and OCO-2 L2Lite Nadir observations for the Tsukuba, Saga, Hefei, Xianghe and Rikubetsu site, respectively, with data screening conditions as defined in Table 4. Panel (a-2)-(e-2) present the boxplots depicting the differences ($\Delta XCO_2$) between the MLP-$XCO_2$ model and OCO-2 products in comparison to the TCCON results for each year. The boxes showing the middle half of the data, from the 25% to the 75% percentiles. The median (50%) is represented by the line within each box. The whiskers encompass the central 90% of the data, extending from the 5% to the 95% percentiles.

compared to traditional methods. While machine learning models need significant upfront time for training data generation and hyperparameter tuning, the prediction is extremely fast once deployed. This enables near real-time processing ideal for operational satellite data streams. Furthermore, the precision and efficiency of neural networks make them well-suited to meet future demands of high-resolution global greenhouse gas monitoring, enabling millisecond-scale $XCO_2$ retrievals suitable for large-scale satellite analysis.

# 6 Conclusions

This proof-of-concept study aims to use the efficient regression inversion capability of the machine learning method to develop machine learning models based on simulated atmospheric radiative transfer data for efficient inversion of satellite observed spectra to retrieve $XCO_2$. This helps overcome the low efficiency in traditional optimization-based iterative algorithms for $XCO_2$ retrievals. In the presented study, $XCO_2$ inversion models using both satellite product based and simulation based data were developed, trained and tested. Long-time series inversion and prediction of OCO-2 observations over East Asia were also performed using the developed models. The results were compared with OCO-2 and TCCON retrievals, showing the simulation data based machine learning models can effectively eliminate lagging biases while achieving millisecond-level (<1 ms) inversion efficiency, good accuracy (less than 1.8 ppm), local emission source capture, and long-term prediction stability. It should be noted that our current MLP-$XCO_2$ model does not provide direct uncertainty estimates, estimating prediction intervals is an important next step for future improvements. Additionally, to provide good prior information while preventing the model from potentially focusing solely on interpolation rather than learning about actual $CO_2$ increases within spectra, our investigation has suggested that integrating additional contextual information, such as the "year," can offer valuable context for $XCO_2$ retrieval. However, the underlying mechanisms behind this improvement may require further investigation.

*Code availability.* The ReFRACtor model and its OCO retrieval implementation can be accessed from the Github ReFRACtor repository (https://github.com/ReFRACtor, last accessed in August 2023). The codes and models used in this study have been uploaded to GitHub and can be accessed at: https://github.com/TaoRen-Rad/XCO2_retrieval. For access to the dataset, please send requests to Tao Ren (tao.ren@sjtu.edu.cn).

*Data availability.* The OCO-2 products (including OCO-2 L1B, Met, L2std and L2Lite files) are available from Goddard Earth Sciences Data and Information Services Center (https://disc.gsfc.nasa.gov/datasets/, last access: March 2023). The TCCON site products are available from TCCON DATA ACHIEVE (https://tccondata.org/, last access: March 2023).

*Financial support.* This research has been supported by the National Natural Science Foundation of China (Grants Nos. 52276077 and 52120105009)

*Author contributions.* FX and TR designed the study. FX made updates and modifications for the Refractor forward model and developed the machine learning code, carried out the tests and result analysis under the supervision of TR. FX and TR prepared the manuscript. All authors reviewed the manuscript.

*Competing interests.* The corresponding author has declared that none of the authors has any competing interests.

*Disclaimer.* Publisher's note: Copernicus Publications remains neutral with regard to jurisdictional claims in published maps and institutional affiliations.

*Acknowledgements.* We gratefully acknowledge the TCCON Data Archive hosted by CaltechDATA at https://tccondata.org for providing the TCCON data for our study. Our thanks also go to the OCO-2 Science Team for the OCO-2 project. We appreciate our colleagues for their feedback and our affiliated institutions for their unwavering support.

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
