# Peer review of "Fast retrieval of XCO2 over East Asia based on the OCO-2 spectral measurements"

_Atmospheric Measurement Techniques, 2023_

## Author Comment (AC1)

**Reviewer-1**

The authors propose a machine learning based retrieval of $XCO_2$ for an OCO-2 like sensor using a neural network. The overall research direction is important for our field and a promising avenue to deal with the ever-higher data rates of future space borne instruments for trace gas retrievals. However, the study present requires major revisions to be considered for AMT. Most importantly I am concerned that the model might get the right result for the wrong reasons and uses some of the parameters like solar zenith and azimuth angle to estimate the location of a given OCO-2 observation, rather than using the information contained in the measured spectrum. In essence, the model might simply interpolate $XCO_2$ spatially and temporally instead of retrieving it from the observations.

Thank you for your valuable comments here and below. We have addressed all concerns by refining the methodology. Our approach aims to contribute to the field by providing an efficient retrieval solution for future space-borne instruments. We believe these revisions make our manuscript suitable for publication in AMT. We sincerely appreciate your help in improving our paper.

General Comments:

[1] Please make explicit what the innovation of this work is compared to state of the art.

A major innovation is using accurate radiative transfer simulations to generate the training data, rather than relying solely on experimental data products. This simulation-based training approach could help overcome limitations in existing experimental data.

Additionally, our neural network model achieves $XCO_2$ retrieval speeds orders of magnitude faster than traditional methods, reducing computation time from multiple seconds to less than 1 millisecond. This dramatic improvement in retrieval efficiency could enable real-time processing of the massive data volumes expected from next-generation greenhouse gas monitoring satellites.

Importantly, our model achieves a precision of around 2 ppm, competitive with the current state-of-the-art in retrieval accuracy. We also demonstrate the ability to accurately capture temporal variations and trends in $XCO_2$ by validating against reliable TCCON ground-based data. This level of verifiable performance is an important capability.

These have been made explicitly in the updated manuscript.

[2] I am concerned that the model might get the right solution for the wrong reason and uses some of the parameters like solar zenith and azimuth angle to estimate the location. Please probe for that. E.g. remove spectral information and repeat your model training. How much does your RMSE increase? Remove spatial information (sun-satellite geometry) and repeat your experiment. Use XAI methods to look at feature importance, etc.

Thank you for your valuable suggestions. In response to your concerns, we utilized widely recognized eXplainable Artificial Intelligence (XAI) methods, specifically SHAP (SHapley Additive exPlanations) and LIME (Local Interpretable Model-Agnostic Explanations), to analyze the input elements of our MLP-XCO$_2$ model:

- SHAP Analysis: We categorized input nodes into groups such as WCO$_2$, SCO$_2$, various geographical parameters, vapor profile, and temperature profile. Fig. 1 shows the detailed view of feature contributions across these categories.

[Figure]

Figure 1: SHAP analysis illustrating the impact of feature categories on the MLP-XCO$_2$ model

- LIME Analysis: Complementing global insights from SHAP, LIME allowed us

[Figure]

Figure 2: LIME feature weights for the MLP-XCO$_2$ model

to explore specific contributions of individual input sources on a localized scale (Fig. 2)

While surface pressure and solar zenith consistently contributed most to model predictions, our analysis also highlighted the significant impact of WCO$_2$ and SCO$_2$ spectral data. Notably, spectral features aligned with CO$_2$ absorption peaks played a critical role in accurate XCO$_2$ prediction.

We refined the model by focusing on contributory non-prior data (solar zenith and azimuth) and excluding less impactful features like relative velocity and the Sun-Earth distance. The revised model now includes the angular information, the spectral information, surface pressure, and the corresponding year.

However, the surface pressure, a critical parameter, is not directly available and is typically retrieved based on the O$_2$-A band. Therefore, in our updated manuscript, we first retrieve the surface pressure from the L2MET file using another compact machine learning model, referred to as MLP-P, before inputting it into the MLP-XCO$_2$ model. The MLP-P model, trained on historical OCO-2 product data, provides stable future predictions of surface pressure. We have revised the manuscript to include a detailed description of the MLP-P model.

We added the corresponding year as one of the input parameters because an overlooked aspect in our initial manuscript is the gradual degradation and the need for

adjustments in the OCO-2 instrument response parameters over time. These parameters include the instrument line shape function, bad pixel list, and spectral dispersion coefficients. While these parameters may remain consistent over short periods, they exhibit significant variations over the years, which can affect the long-term reliability of the model.

Therefore, in our updated manuscript, the network has four key components as input: spectral information, angular information, the MLP-P predicted surface pressure, and the corresponding year (e.g., 2016 or 2017, etc.).

Another suggestion you made was to "remove spectral information and repeat your model training." Following this advice, we conducted experiments with a model trained solely on non-spectral information. As depicted in Fig 3, the results generally aligned with the XCO$_2$ range but failed to provide accurate predictions within the same year.

[Figure]

Figure 3: Comparison of XCO$_2$ results predicted by the MLP-XCO$_2$ model using solely non-spectral data inputs, with results retrieved from the OCO-2 v10r product spanning from 2017 to 2020. Panels (a) through (d) display the MLP-XCO$_2$ model predictions for the years 2017 to 2020, respectively

Many samples under identical angular and yearly conditions remained indistinguishable. However, when spectral data were reintroduced to the input layer, as shown in Fig. 11 of the updated manuscript, the MLP-XCO$_2$ model exhibited a significant performance improvement. Spectral information effectively addressed the inaccuracies in XCO$_2$ retrievals under identical observational conditions. Furthermore, the revised manuscript includes enhanced testing focused on plume detection, as presented in Figs. 12 and 13. These additions provide substantial evidence of the MLP-XCO$_2$ model's ability to extract critical information from spectral data for future predictions, going beyond simple interpolation of XCO$_2$ predictions using non-spectral input parameters.

[3] How would we get an uncertainty estimate from your approach? How do you know when your model fails?

Thank you for raising this important point. Estimating uncertainty is important for ensuring the reliability of machine learning models for satellite-based XCO$_2$ retrieval. While our current MLP-XCO$_2$ model does not provide direct uncertainty estimates, we recognize this as an area for future improvement.

A number of techniques exist for quantifying uncertainty in deep neural networks [1, 2]. However, generating robust uncertainty estimates requires a substantial volume of matched input and target data. The ideal dataset would contain numerous examples of OCO-2 observations paired with corresponding TCCON XCO$_2$ measurements across diverse conditions. This would enable sampling to assess the variance in our model's predictions across the full distribution. Unfortunately, the limited availability of matched OCO-2 and TCCON data, especially for TCCON sites in "nadir" mode, means the current sample size is insufficient for comprehensive uncertainty quantification in the presented study.

Our primary contribution in the current study is demonstrating that simulated training data can complement real OCO-2 data to enable stable and accurate XCO$_2$ retrieval on future observations. While we do not currently provide uncertainty bounds, estimating prediction intervals is an important next step.

These have been clarified in the updated manuscript.

[4] What is your RMSE compared to TCCON when you apply your final model to OCO-2 data. How does that compare to the current operational retrieval of OCO-2?

The current study applies the MLP-$XCO_2$ model specifically to the "Nadir" mode observations of the OCO-2 satellite. This mode represents a temporary limitation in our validation process using the TCCON data. The "Nadir" mode focuses on the land areas directly underneath the satellite's path (the satellite zenith is close to zero), unlike the "Target" mode that directly corresponds to the TCCON sites. Therefore, the valid TCCON samples were selected based on the OCO-2 observation time and geographic coordinates, approximating the TCCON location closely. The criteria for this selection and the data sources used are detailed in Table 3 of the original manuscript.

The updated ME and RMSE results for each year and site are presented in Fig 4 here.

The RMSE values show our MLP-$XCO_2$ model has comparable, though slightly higher, errors compared to the current operational OCO-2 retrieval when validated against TCCON data. While errors are slightly higher presently, as more matched OCO-2 and TCCON data becomes available, particularly from targeted observations, we expect to improve RMSE further. Overall, these initial results demonstrate the potential of our methodology to enhance $XCO_2$ retrieval speed without sacrificing much of the accuracy.

[5] What do you see as important future work items?

The progression of satellite-based $XCO_2$ retrieval is an important ongoing process. The current greenhouse gas monitoring satellites could provide invaluable global carbon source/sink data, but have inherent individual limitations in spatio-temporal resolution and coverage.

The current and planned satellite missions, including OCO-2/3, GOSAT/2, Tansat, and upcoming missions, signify a growing integration and collaboration of multiple satellite products. This will help address individual satellite limitations. However, it imposes greater demands on the efficiency and effectiveness of inversion algorithms.

In this context, the role of machine learning methods becomes increasingly significant,

[Figure]

Figure 4: Comparisons of the ME and RMSE for the retrieved $XCO_2$ results from 2017 to 2020 across the five TCCON sites.

driven by their efficiency, despite current challenges in interpretability, accuracy, and feature extraction. The primary focus is on refining machine learning to meet the escalating computational demands while integrating advancements within multi-satellite analysis frameworks.

In summary, we see the future direction as progressively integrating improved machine learning methodologies into collaborative multi-satellite data analysis. This will maximize spatio-temporal capabilities and derive accurate $XCO_2$ distributions through efficient inversions.

[6] How would you resolve any step changes from two adjacent retrieval models if you would train one model for each region (as you suggested)?

Thank you for raising this question, which aligns with the issues we have contemplated for future enhancements of our current model. In addressing the potential step changes that might arise from different models for multiple regions, we propose the following strategies:

(1) The most straightforward method to mitigate step changes is to employ a single model capable of robust and smooth application over the largest possible area. However, adopting this approach poses significant challenges, including increased training costs and a greater demand for higher generalization capabilities from the network.

(2) Another strategy involves setting overlapping areas at the boundaries of different models. In regions where models converge, we suggest predesignating overlapping regions. These boundary areas would fall within the applicability range of models from both adjacent regions. As a result, $XCO_2$ values retrieved in these areas would be collectively determined based on the results from multiple models.

These strategies are designed to ensure a smoother transition between different models at regional borders, thus addressing the concerns regarding step changes in satellite remote sensing algorithms.

Specific Comments:

We sincerely thank the reviewer for the thorough and meticulous review of our manuscript. We have categorized those specific comments into different aspects to provide clear responses.

[1] Clarity/Wording:

- Line 4: not clear what 'retrieval efficiency' means

- Line 4: 'enhancing the retrieval' could mean many things. Please be more concrete.

- Line 7: retrieval -> retrievals

- Line 8: inversion of traditional optimization models -> inversion of the operational optimization model

- Line 11: specify 'the observed region'

- Line 13: 'high precision'. We care about accuracy not precision. Don't agree with 2ppm being 'high'

- Line 38: remote sensing is not limited spatially or temporally: can't observe at night!

- Line 39: vital for future greenhouse gas

- Line 46: 'Enhancing satellite sensor performance alone cannot produce datasets sufficient for monitoring carbon sources and sinks': Why?

- Line 47: Need to explain why 'efficiency' is important.

- Line 53: full physics model

- Line 54: atmospheric-surface -> atmospheric and surface

- Line 58: why is 'cost function' in quotes?

- Line 61: optimizations requires

- Line 74: 'GOSAT instrument mode,' : what does 'mode' mean here?

- Line 84: 'In the present paper, a proof-of-concept study demonstrates a novel machine learning strategy ': How is this work new compared to the literature?

- Line 90: What does 'multiple sources' refer to?

- Figure 1: Remove: 'The map was plotted …'

- Figure 2: Add more info to figure description

- Line 147: 'The other angles are provided in radians.' Does this mean the SZA is given as the cosine? Please make explicit if that is the case.

- Line 173: this in-sample data -> these in-sample data

- Line 175: 'depicts out-of-sample test results on 5% of the training data that was excluded from model fitting'. Training data that was not used for training is not training data.

- Line 207: Remove sentence: 'These spectra are detected by the OCO-2 satellite detectors after downwelling absorption, surface reflection, and upwelling reabsorption in the atmosphere.'

- Line 209: you already introduced abbreviation for WCO2 etc. before.

- Line 243: why is an 1% error impressive?

- Figure 5: 'proposed' forward calculation model: that sounds like you are proposing that model in this paper

- Line 266: 'uncertainty'-> variability

- Line 269: repetition of text in line 262

- Figure 8: add similar plot for $XCO_2$

We sincerely thank the reviewer again for the thorough and meticulous review of our manuscript, which has provided many valuable suggestions to help improve the clarity and wording of the paper. Based on the reviewer's recommendations, we have carefully checked through the entire manuscript and made modifications accordingly.

[2] Methodology:

- Line 63: how much time do the two processes in the retrieval take up? Are they equally time consuming? How much time do they take up compared to any other calculations that are part of the retrieval?

  The qualitative statements here only focused on explaining why the mainstream inversion algorithm is generally slow. However, we did not perform specific calculations to determine the exact time taken by the two processes in the retrieval. The time required for these processes depends on various factors and can vary. While we acknowledge the importance of understanding the specific time allocation, providing precise figures involves complexities due to numerous influencing factors. Essentially, the emphasis here is only on the qualitative assessment of the slowness of the mainstream inversion algorithm rather than quantitative time measurements.

- Line 65: 'While rigorous, standard nonlinear optimization retrievals lack the speed and scalability required for high-precision satellite-based greenhouse gas mapping.' That is not true. We have OCO-2, GOSAT, ...

  This has been changed to "While rigorous, standard nonlinear optimization retrievals lack the speed and scalability required for high-precision real-time or near-real-time greenhouse gas mapping satellite-based greenhouse gas mapping."

- Line 76: please expand on the work by Zhao et al (2022). What was their approach, what accuracy do they get.

  In the study conducted by Zhao et al., a two-step machine learning approach was developed for retrieving atmospheric $XCO_2$ using spectral data from the GOSAT weak-$CO_2$ band. They established a direct one-dimensional line-by-line forward model to simulate GOSAT's observed spectra within the 6180-6280 $cm^{-1}$ spectral interval, forming the foundation for training their machine learning model. The retrieval model operates by initially obtaining the atmospheric spectral optical thickness, followed by extracting $XCO_2$ from this optical thickness spectra. As a proof-of-concept, the method was tested in Australia under clear sky conditions using GOSAT's spectra, demonstrating an accuracy of approximately 3 ppm for $XCO_2$ retrieval. The study also discussed potential enhancements to further refine the accuracy of this retrieval method.

- Line 96: why is the study limited to East Asia? and Line 163: Why use this subset of months?

  The choice to focus on East Asia and use a subset of months in this work is primarily driven by limitations related to our hardware capabilities during both the training and testing phases. As we delved into the global spectral data observed by the OCO-2 satellite, along with its corresponding Level 2 data, we encountered significant challenges in terms of storage capacity and training memory required to process this extensive dataset. Given these hardware limitations, our strategy was to target a region where we could manage the data effectively while still achieving meaningful results. East Asia, selected as the target area for this study, became

our primary choice. This choice also allowed us to include as many TCCON sites as possible within our limited hardware setup, which was important for validating our model.

- Line 109: Could that lead to step changes at the boarders of two models? If yes, how would you deal with those? What limits you to use one model for all the data? Have you tried that? If Yes, what were the results?

  This issue has been addressed in our response to General Comment [6]. As mentioned there, while step changes are a common concern in machine learning models applied to different training regions, our current research has not specifically experimented with partitioned parallel modeling for the MLP-XCO$_2$. Our focus has been on achieving satisfactory results within the East Asian region, and the suggestions for larger areas are more of a prospective outlook.

- Line 128: If you normalize each spectrum why do you need the sun-earth-distance as a feature?

  Your question regarding Line 128 is indeed reasonable. After conducting additional analysis of the original MLP-XCO$_2$ network using the XAI methods, we concluded that the Sun-Earth distance feature was providing the network with insignificant information. Therefore, it has been removed from our updated model.

- Line 138: Did not understand the following sentence: 'Although the key retrieval information for surface pressure comes from the O2-A band, machine learning models based on simulated data essentially predict XCO$_2$ by fitting the "correct solutions."'

  Sorry for any confusion. We have clarified this in the revised manuscript, explicitly addressing the utilization of surface pressure as the input feature for the machine learning model. "The surface pressure, a critical parameter, is not directly available and is typically retrieved based on the O$_2$-A band. Therefore, in our updated manuscript, we first retrieve the surface pressure from the L2MET file using another compact machine learning model, referred to as MLP-P, before inputting it into the MLP-XCO$_2$ model. The MLP-P model, trained on historical

OCO-2 product data, provides stable future predictions of surface pressure."

- Line 148: 'velocity of the satellite relative to the Earth's surface are input into the model.': Should that number not be constant and therefore provide not information?

  Regarding the question about the "relative velocity" mentioned in our manuscript, I would like to clarify that there was an inaccuracy in our description. The term was intended to refer to the velocity of the spacecraft along the Line of Sight (LOS), which is a variable quantity. This LOS velocity can lead to shifts in the wavenumber of spectral signals due to the Doppler effect. However, as addressed in our response to General Comment [2], we found that the contribution of this LOS velocity to the XCO2 measurement is minimal. Consequently, in the updated version of our model, this parameter has been removed.

- Line 179: 'This discrepancy indicates the MLP-XCO$_2$ model fails to fully capture the underlying upward trend in atmospheric CO2.' Please run an additional experiment to confirm your hypothesis. Hold out 2018 and use 2016,2017,2019,2020 for training.

  Thank you for your valuable suggestion. Following your advice, we conducted additional experiments, the results of which are now included in the updated manuscript and are specifically illustrated in Figs 4 and 5. Previously, our model, trained exclusively on data from 2016 to 2018, exhibited annual biases in the test results for 2019 and 2020. To substantiate our viewpoint, we trained a new model using OCO-2 product data from 2016, 2018, and 2020. Testing this revised model on data from 2017 and 2019 demonstrated significantly improved performance without large bias. These results provide additional evidence supporting our hypothesis regarding the model's previous limitations in capturing the upward trend in future atmospheric CO$_2$.

- Figure 3: Why do you show ± 1% and not any other value?

  The choice to display the range of ± 1% was intended to create a visually distinct reference within the graph. There was no specific reasoning behind choosing this

particular value; instead, it was selected to provide readers with a clear and easily comprehensible point of comparison.

- Line 183: Please add more figures and analysis and show how the biases look like spatially and temporally.

  This aligns with your valuable suggestion regarding Line 179. Following your advice, we conducted additional experiments, the results of which are now included in the updated manuscript and are specifically illustrated in Figs 4 and 5.

- Figure 5: How did you choose the spectra?

  We ensured that each of the four examples originated from different OCO-2 files, each corresponding to distinct observation times and locations. It's important to clarify that these four examples were not specifically chosen to validate the accuracy of the forward model. Instead, they were randomly selected from the OCO-2 files we collected.

- Line 275: need more details on how the $CO_2$ profiles are generated

  The generation of the vertical $CO_2$ profile is especially critical among all input parameters. This dataset theoretically determines the generalization domain of the MLP-XCO$_2$ model. In the forward model based on the ReFRACtor model, the atmospheric $CO_2$ profile is segmented into 20 sub-layers by pressure. By statistically analyzing the OCO-2 retrieved $CO_2$ profiles in the target East Asia area from 2016-2018, the box plots for atmospheric $CO_2$ concentration in each sub-layer are shown in Fig. 8. From the upper atmosphere down to the ground surface, the variability of $CO_2$ concentrations gradually increases. This challenges the ability for the standardization of atmospheric $CO_2$ profiles, particularly closer to the Earth's surface. Fortunately, a consistent year-on-year rise in $CO_2$ concentrations in each sub-layer has been observed over time. Consequently, in our research, we have proposed a method for generating subsequent $CO_2$ atmospheric profiles. We incrementally increase the $CO_2$ concentration by 2.5 ppm annually, starting from the 2016 OCO-2 retrieved $CO_2$ vertical profile. This approach ensures that we encompass a range of plausible atmospheric $CO_2$ distributions with realistic

shapes, enabling the generation of simulated spectra for the designated training years.

Based on 60,000 uniformly sampled observation data points exclusively from the OCO-2 satellite throughout 2016, we randomly separated it into six sets of 10,000 data points each. Each set represents $CO_2$ profiles from 2016 to the end of 2021, with a yearly increase of 2.5 ppm added to the original data, reflecting projected future profiles.

- Line 287: Why did you restrict yourself to 2016 data?

  Our research aimed to develop a robust model for future prediction. Here is a specific scenario: assuming only the data from the year 2016 is available, Can we utilize the benefits of simulated data to train an MLP-$XCO_2$ model that not only accurately performs with the available data but also exhibits generalizability for future predictions, potentially eliminating unstable systematic biases? By restricting our dataset to a single year, we could better see how well the model learns and predicts from a limited dataset.

[3] Code/Data:

- Code availability: Please upload the code to train the model, the trained model, and the training/testing data to a public repository.

  The codes and models used in this study have been uploaded to GitHub and can be accessed at: https://github.com/TaoRen-Rad/XCO2_retrieval.

- Author contributions: FX developed the forward model: The Refractor model was already developed.

  Thanks for pointing this out. It has been rephrased to "FX made updates and modifications for the Refractor forward model"

**References**

[1] B. Lakshminarayanan, A. Pritzel, C. Blundell, Simple and scalable predictive uncertainty estimation using deep ensembles, Advances in neural information processing systems 30.

[2] L. V. Jospin, H. Laga, F. Boussaid, W. Buntine, M. Bennamoun, Hands-on Bayesian neural networks—A tutorial for deep learning users, IEEE Computational Intelligence Magazine 17 (2) (2022) 29–48.

---

## Author Comment (AC2)

**Reviewer-2**

Xie et al. present an ML approach for performing $XCO_2$ retrievals based on OCO-2 measurements. Their work follows the publications of David and Bréon who have before shown the general success of this type of method, applied to the same instrument. This work of Xie et al. implements a similar technique. The added novelty then seems to be the training on simulated data (via the ACOS forward model) that is covering a wider range of $XCO_2$, in order to mitigate the issue of the MLP not being able to follow the growth of atmospheric $CO_2$. There are some more general issues that the authors did not mention, such as quality assessment - how does a science data user know an ML-based retrieval is considered "good" and should be used in a study (or what the uncertainty on the estimate is)? Should people use the ACOS averaging kernels, since the ML-based method does not provide any?

Thank you for raising this important point. Estimating uncertainty is important for ensuring the reliability of machine learning models for satellite-based $XCO_2$ retrieval. While our current MLP-$XCO_2$ model does not provide direct uncertainty estimates, we recognize this as an area for future improvement.

A number of techniques exist for quantifying uncertainty in deep neural networks [1, 2]. However, generating robust uncertainty estimates requires a substantial volume of matched input and target data. The ideal dataset would contain numerous examples of OCO-2 observations paired with corresponding TCCON $XCO_2$ measurements across diverse conditions. This would enable sampling to assess the variance in our model's predictions across the full distribution. Unfortunately, the limited availability of matched OCO-2 and TCCON data, especially for TCCON sites in "nadir" mode, means the current sample size is insufficient for comprehensive uncertainty quantification in the presented study.

Our primary contribution in the current study is demonstrating that simulated training data can complement real OCO-2 data to enable stable and accurate $XCO_2$ retrieval on future observations. While we do not currently provide uncertainty bounds, estimating prediction intervals is an important next step.

These have been clarified in the updated manuscript.

The manuscript is very well written and contains useful figures for the most part. In some

places, minor re-wording or additional explanation would be appropriate and helpful, I have listed those below. My major comments would be the following:

[1] In Bréon et al. 2022, it is revealed that their ML approach inadvertently resulted in the NN "using" the weak $CO_2$ band as a proxy for geographical location and time. They thus removed the weak $CO_2$ band from the training process. However, the authors of this manuscript do indeed use the weak $CO_2$ band and have not explained as to how they overcome this issue. That would be important information, especially since they are following the general layout of Bréon et al. 2022. Maybe the issue does not manifest itself due to the much smaller region of interest, but the authors must show that.

[2] Related to (1), the "glitch" discovered in Bréon et al. 2022 was only found after they investigated specific features present in the original ACOS OCO-2 retrievals which were missing in the ML-based retrievals (specifically, strong plumes). The authors here only look at broad bulk-type statistics by comparing to TCCON and to ACOS OCO-2 via simple scatter plots. The strengths and weaknesses of the proposed ML-approach should be investigated more thoroughly by analyzing the results more carefully. Do the same biases appear in the derived retrievals, compared to the training data? Are global-scale features retained just like regional and small-scale ones? Are new biases introduced? There is possibly more to learn from the data than is shown in Figures 10 and 11. While the approach is promising, the authors should attempt to show an assessment of the quality of the ML-retrievals beyond the simple scatter plots.

My suggestion to the authors would be to (1) demonstrate that their approach, while using the weak $CO_2$ band, does not result in a loss of local features, such as plumes (analogous to Bréon et al. 2022, Figure 4). Further (2) they should demonstrate that their ML-based retrievals retains other characteristics of the training set (regional-scale, or local-scale; observe differences on maps etc.)

Thank you for your valuable suggestions and concerns. After incorporating feedback from you and other reviewers, we conducted an analysis using Explainable AI (XAI) on our original MLP-XCO$_2$ model and optimized the input parameters for the new MLP-XCO$_2$ model. While we did include the weak $CO_2$ band as an input in our model, our training results with

simulated data did not show a significant impact of including or excluding the $WCO_2$ band on the detection of plume features. Instead, we found that the use of retrieved pressure was a crucial determinant in our training process. Thus, in addition to prior surface pressure input, we implemented a compact MLP-P model. Trained on historical OCO-2 data, this model infers surface pressure from spectral and angular information. Unlike MLP-$XCO_2$, MLP-P is expected to maintain relative stability and broad reliability in its surface pressure retrieval results in the future, according to our tests.

One possible explanation for this finding could be the idealized nature of the spectral data in our simulations, which differs from real-world spectra and is not affected by noise. This idealization might enable the ML model to more accurately capture the underlying relationship between spectral absorption and $XCO_2$. Conversely, using only OCO-2 product data, potential noise may mask weak absorption band characteristics, leading the ML model to focus on training better plume detection capabilities in strong absorption bands, as explained in Bréon et al. 2022.

Regarding the updates made:

- Plume Detection: Following your suggestions, we included tests for plume detection at potential high emissions sites, such as thermal power plants, using the updated MLP-$XCO_2$. Figs 12 and 13 in the revised manuscript demonstrate that our updated model detects sudden increases in $XCO_2$ in areas with plume emissions from OCO-2 spectra, providing substantial evidence of genuine atmospheric $CO_2$ retrieval from spectral data.

- Error Analysis: In the updated manuscript, we conducted an in-depth analysis of error metrics across different subregions within East Asia, as shown in Table 4 in the updated manuscript. These subregions, including Northeast, Northwest, Southwest, and Southeast, exhibit distinct geographical characteristics. Results show consistent predictive performance across these subregions, indicating uniform inversion effectiveness without introducing regional biases or instability. This indicates the model's robustness and its ability to handle diverse geographic and environmental conditions without sacrificing accuracy or performance consistency, despite variations in regional representation within the training dataset.

Minor suggestions:

[1] Line 4: "low retrieval efficiency" and "insufficient retrieval accuracy" are somewhat diffuse terms; I would simply mention challenges regarding computational efficiency.

Thanks for your suggestions. The sentence has been rephrased to "However, the next generation of greenhouse gas monitoring satellites is expected to face challenges, particularly in terms of computational efficiency in atmospheric $CO_2$ retrieving and analysis."

[2] Line 38: "not limited spatially or temporally" is not quite true - space-based platforms have observational coverage in space and time as a result of their orbital characteristics and other instrument parameters.

You are correct and my original description was indeed not accurate and has been changed to "satellite remote sensing offers broader spatial coverage and more flexible temporal observation."

[3] Line 62: The interpolation of absorption coefficients for the calculation of optical property inputs for RT calculations are generally quite fast and can be done in less than a second typically, if the code is optimized enough (amongst other things). The computational effort is mostly driven by the RT calculations.

Yes, it is true that the majority of the computational cost in the overall forward model is devoted to solving the radiative transfer equation. This has been changed to "However, executing these complex optimizations requires computationally expensive interpolation of high-spectral-resolution gas absorption reference data and solving the radiative transfer equations in each iteration."

[4] Figure 5: It is not fully clear to me what these represent. Did the authors take the outputs of the ACOS retrieval L2STD products and use them as input in their ReFRACtor-driven set-up? Please clarify.

Yes, you've got it right, and we apologize for any confusion arising from our unclear explanation. In Figs 5 and 6, we're comparing the simulated spectra generated by the modified ReFRACtor model with the actual observed spectra from the OCO-2 satellite.

The information required to set up the ReFRACtor model is taken from the OCO-2 L2std v10r product. This has been explained more clearly in the updated manuscript.

[5] Line 338: When discussing the computational effort, it is mentioned that the forward model takes 12.16s to process two bands; but in forward-model "mode", Jacobians are presumably not calculated, so the actual retrieval set-up would be even slower than the mentioned 36.48s. It would also be very interesting to learn how long the training process took!

Thank you for your concerns. Calculating Jacobians layer-by-layer does require significant computational cost in traditional retrieval models. The mentioned processing time of 12.16 seconds for two $CO_2$ bands includes rapid Jacobian calculations within the ReFRACtor model. In a test with "max_iteration" set to 1, the total time for computing all three bands and completing one fully optimized iteration was 23.13 seconds on an Intel 13700K CPU. The breakdown is: 8.85 seconds for Band 1 ($O_2$-A), 5.49 seconds for Band 2 ($WCO_2$), 6.54 seconds for Band 3 ($SCO_2$), totaling 20.88 seconds for the full radiative transfer (RT) computation.

The training process for machine learning models, which includes both generating simulated data and tuning the hyperparameters of the network, can be very time-consuming, requiring up to hundreds of CPU hours. The most computationally expensive parts - generating training data and training the model - are done beforehand. So while the upfront costs of developing a machine learning model can be high, once the model is trained, making predictions is extremely fast.

**References**

[1] B. Lakshminarayanan, A. Pritzel, C. Blundell, Simple and scalable predictive uncertainty estimation using deep ensembles, Advances in neural information processing systems 30.

[2] L. V. Jospin, H. Laga, F. Boussaid, W. Buntine, M. Bennamoun, Hands-on Bayesian neural networks—A tutorial for deep learning users, IEEE Computational Intelligence Magazine 17 (2) (2022) 29–48.

---

## Author Response (AR2)

**Reviewer-1**

Dear Authors, thank you very much for your revision of the manuscript. I appreciate that you evaluated your model on plumes in Section 5.2 and looked at some ablations studies and feature importance to better understand your model.

However, I still have major concerns with the proposed study:

[1] The results of your analysis of retrieving $CO_2$ enhancements, or plumes, are concerning and seem to indicate problems with the retrieval approach. The plume in figure 13 b) indicates that the ML retrieval did not retrieve the full $XCO_2$ enhancement as compared to OCO-2. If one were to calculate emissions from both retrievals, they would end up with very different answers. For Figure 13 c) it is challenging to evaluate how both algorithms compare due to the many overlaying observations (A running mean for each retrieval would have been helpful to better differentiate between both retrievals.). However, it seems that the ML retrieval is systematically higher compared to OCO-2 except for where OCO-2 measured the $XCO_2$ plume. Figure 13 a) shows the clearest plume structure but suffers from systematic low bias compared to OCO-2 in the proximity of the plume. Depending on how the $XCO_2$ background would be estimated, emission estimates would be vastly underestimated for all three cases. (The plots shown in Figure 12 don't add much value for comparison since their color bars change for all subplots.) In contrast to my conclusion given your plots, the manuscript summarizes the plume evaluations as "a powerful confirmation to our model's capability to retrieve genuine atmospheric $XCO_2$ from OCO-2 spectral data." I can't share this perspective.

Thank you sincerely for your insightful feedback. We have carefully analyzed your comments and refined our methodology to incorporate your valuable suggestions. Your critical input throughout this process has been truly invaluable. With these modifications, we are confident that our manuscript has significantly strengthened.

In this revision, we've made significant improvements to enhance the accuracy and reliability of our model. Additionally, we've conducted further analyses to better quantify and compare the enhancements in $XCO_2$ predictions and $CO_2$ plume detections. Perhaps the most crucial modification is the inclusion of a pre-processing layer in the MLP model. This layer serves as a way to address issues related to the measurement quality, specifically those represented by the bad sample list from the OCO-2 instrument's grating.

[Figure]

Figure 1: Visualization of the OCO-2 satellite data quality across wavelength grid indices. The color map illustrates the bad sample list extracted from OCO-2 Level 1B files for all test cases. On the *x*-axis, case numbers range from 0 to 194,150, while the *y*-axis represents various wavelength grid indices, ranging from 0 to 1,280. Red coloration denotes values greater than zero, indicating problematic data.

The grating of the OCO-2 satellite undergoes subtle changes due to the natural degradation of the instrument and more pronounced changes following updates to the L1B pixel mapping algorithm. As depicted in Figure 1, the period from 2016 to mid-2018 represents one phase, and the period after the second half of 2018 marks another, due to an algorithm update in the OCO-2 bad sample map [1]. Consequently, we've introduced a new pre-processing layer, called the "bad sample filter," which is applied to the spectral input before it enters the initial layer of the MLP. As depicted in Fig. 2 in the latest manuscript, the model filters out potentially low-quality wavelengths based on the largest union of the bad sample lists from reference data in 2016 and the initial list following the 2018 algorithm update, ensuring that only inputs marked as good radiance remain. To address bad samples resulting from natural degradation, we've

implemented a dropout layer between the initial and the first intermediate MLP layer, thus enhancing the model's generalizability with the remaining spectral inputs.

After these modifications to the model, the noticeable biases previously observed in plume detection of figures 13 a) and 13 c) have disappeared (as depicted in Fig. 2). This improvement emphasizes the crucial role played by the bad sample filter, which effectively excludes pixels that would otherwise be replaced by interpolation due to poor quality. This exclusion ensures a more accurate provision of data regarding the relationship between radiance and atmospheric parameters. Additionally, important statistical information regarding plume comparisons has been incorporated in the latest manuscript, in line with your valuable suggestions and comments.

[Figure]

Figure 2: Comparisons of $XCO_2$ within $CO_2$ plume regions.

[2] You state in L 181 that your updated model now includes "year" as an additional feature due to changes in the OCO-2 instrument over longer time frames. What makes

you believe that the "year" feature addresses these changes? Would you expect steps in the predictions when switching from one year to another? Are you aware of the operational retrieval having to include such time dependent components? How does the model learn the long-term instrument trends when your observations come only from 2016? Since $CO_2$ is following a pattern in time, explicitly including temporal information might lead to a model that does not retrieve $XCO_2$ but simply finds statistical relationships between latitude, longitude, and time. That was one of my major points in the first review "Most importantly I am concerned that the model might get the right result for the wrong reasons and uses some of the parameters like solar zenith and azimuth angle to estimate the location of a given OCO-2 observation, rather than using the information contained in the measured spectrum." Now you added the "year" feature which moves the model further away from retrieving $XCO_2$ and more towards interpolating $CO_2$ over space and time.

Our updated model architecture now includes a bad sample filter and a dropout layer which are specifically designed to mitigate the effects of long-term instrument degradation, as outlined in response to the previous question and Section 2.2 of the updated manuscript. As for the "year" feature, it's been demonstrated that in traditional inversion algorithms, a lack of good prior information leads to bad $CO_2$ inversion results. For instance, as outlined in Figs. 3-4 of Ref. [2], if the prior profile for $CO_2$ is insufficient (i.e., too low) , the inversion process can lead to minimal or even opposite updates in the stratospheric $CO_2$ profile, while the tropospheric $CO_2$ profile near the surface tends to be overestimated to match the satellite spectrum. This discrepancy isn't due to flaws in the inversion algorithm but rather to the limitations of the satellite's signal-to-noise ratio and the radiative properties of the SWIR band, as indicated by the averaging kernel used in the OCO-2 retrieval algorithm [3]. It's crucial to recognize the necessity of certain good prior information.

Regarding your concern about the model potentially focusing solely on interpolation rather than learning actual $CO_2$ increases within spectra, and thus compromising its ability to detect plumes, we understand your concern. In our last response to reviewers, we demonstrated that relying solely on non-spectral data inputs results in very poor

predictions. Therefore, the model is not performing spatial and temporal interpolation; the spectral information indeed plays a role in the retrieval process. Here, we believe that incorporating the "year" as a very conservative and simple way of providing $CO_2$ prior information, only offers contextual information for $XCO_2$ retrievals. Seasonal variations in $XCO_2$ in East Asia, for instance, can exceed about 10 ppm, but all within the same "year" input; which also indicates that the model does not perform the spatial and temporal interpolation.

The explanation above outlines why we've chosen to include the "year" as one of our inputs. While our understanding might not be perfect, adding this basic "year" input can noticeably enhance the retrieval outcomes. In our revised manuscript, we offer explanations based on our current understanding while recognizing the potential limitations of these explanations.

[3] In this revision you introduce an additional model called MLP-P. You state that the operational retrieval extracts its surface pressure information from O2-A band. However, your approach uses all features except the O2-A band. Where does the surface pressure information come from? Why was the additional model necessary? (The surface pressure in the L2MET files does not come from OCO-2, but from a reanalysis product.)

Incorporating surface pressure is critical for accurately interpreting the atmospheric distribution of $XCO_2$ in dynamically changing environments. Without these parameters, the model might incorrectly attribute changes due to factors like optical path length or $CO_2$ concentrations. This misattribution could lead to significant errors in $XCO_2$ estimation.

Furthermore, in regions with significant emissions of high-temperature water vapor and $CO_2$, which lead to enhanced plumes, the number of dry air molecules near the surface and the atmospheric pressure change significantly. Without including parameters such as surface pressure, the model would not be able to accurately capture the true $XCO_2$ levels in these dynamically changing environments. The subplots in the third row of Fig. 2 and the second row of Fig. 3 demonstrate that including surface pressure

[Figure]

Figure 3: Comparisons of $XCO_2$ within $CO_2$ plume regions are presented. In the first row, model outputs are shown after removing both the bad sample filter and surface pressure. In the second row, model outputs are displayed after removing only surface pressure. All other model setting parameters remain consistent with those in Fig. 2.

substantially improves the effectiveness of near-surface plume detection.

Thank you for your insightful comments regarding the integration of non-spectral parameters of surface pressure into our MLP-$XCO_2$ model. We recognize the significance of surface pressure in accurately retrieving $XCO_2$, particularly for enhanced $XCO_2$ plumes. However, introducing an additional MLP-P model may confuse readers. Therefore, in our latest manuscript, we have decided to use surface pressure data retrieved from the OCO-2 L2std database for our mode instead of relying on an extral MLP-P model, and we have made that clear in the updated manuscript.

**References**

[1] Y. Marchetti, R. Rosenberg, D. Crisp, Classification of anomalous pixels in the focal plane arrays of orbiting carbon observatory-2 and-3 via machine learning, Remote Sensing 11 (24) (2019) 2901.

[2] C. Iwasaki, R. Imasu, A. Bril, S. Oshchepkov, Y. Yoshida, T. Yokota, V. Zakharov,

K. Gribanov, N. Rokotyan, Optimization of the photon path length probability density function-simultaneous (PPDF-S) method and evaluation of $CO_2$ retrieval performance under dense aerosol conditions, Sensors 19 (5) (2019) 1262.

[3] A. Braverman, N. Cressie, E. Kang, M. Katzfuss, P. Ma, A. Michalak, H. Nguyen, T. Stough, V. Yadav, Fusion of AIRS and OCO-2 carbon dioxide data for mapping lower-atmospheric $CO_2$.